# X-DRIVE: CROSS-MODALITY CONSISTENT MULTI-SENSOR DATA SYNTHESIS FOR DRIVING SCENARIOS

**Yichen Xie**[1*], **Chenfeng Xu**[1*], **Chensheng Peng**[1], **Shuqi Zhao**[1], **Nhat Ho**[2],
**Alexander T. Pham**[3], **Mingyu Ding**[1†], **Masayoshi Tomizuka**[1], **Wei Zhan**[1]
[1]UC Berkeley  [2]UT Austin  [3]Toyota North America

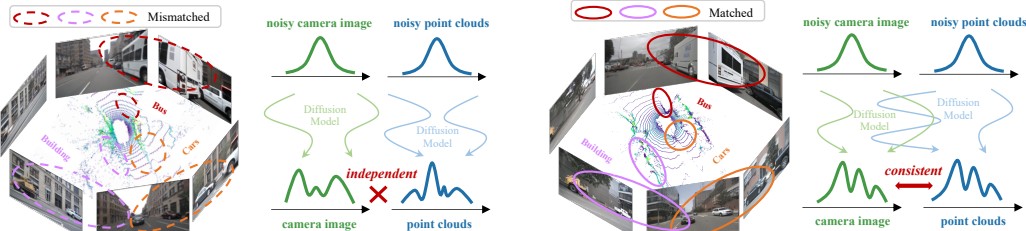

(a) Single-modality images and point clouds synthesized by separate models (Gao et al., 2023; Hu et al., 2024a)

(b) Multi-modality images and point clouds jointly generated by our proposed X-DRIVE.

Figure 1: X-DRIVE simultaneously generates high-quality multi-view images and point clouds with cross-modality consistency, which is impossible for previous single-modality generative models.

## ABSTRACT

Recent advancements have exploited diffusion models for the synthesis of either LiDAR point clouds or camera image data in driving scenarios. Despite their success in modeling single-modality data marginal distribution, there is an under-exploration in the mutual reliance between different modalities to describe complex driving scenes. To fill in this gap, we propose a novel framework, X-DRIVE, to model the joint distribution of point clouds and multi-view images via a dual-branch latent diffusion model architecture. Considering the distinct geometrical spaces of the two modalities, X-DRIVE conditions the synthesis of each modality on the corresponding local regions from the other modality, ensuring better alignment and realism. To further handle the spatial ambiguity during denoising, we design the cross-modality condition module based on epipolar lines to adaptively learn the cross-modality local correspondence. Besides, X-DRIVE allows controllable generation through multi-level input conditions, including text, bounding box, image, and point clouds. Extensive results demonstrate the high-fidelity synthetic results of X-DRIVE for both point clouds and multi-view images, adhering to input conditions while ensuring reliable cross-modality consistency. Our code will be made publicly available at https://github.com/yichen928/X-Drive.

## 1 INTRODUCTION

Autonomous driving vehicles perceive the world with multiple sensors of different kinds, where LiDAR and cameras play crucial roles by capturing point clouds and multi-view images. They provide complementary geometric measurements and semantic information about the surrounding environment, significantly benefiting tasks such as object detection (Liu et al., 2023c; Xie et al., 2023), motion planning (Sobh et al., 2018), scene reconstruction (Huang et al., 2024; Zhou et al., 2024), and self-supervised representation learning (Yang et al., 2024a; Xie et al., 2024b). However, these advancements hinge on access to *large amounts of aligned multi-modality data*, specifically well-calibrated LiDAR and multi-view camera inputs that describe the same scene.

Scaling up the collection of such high-quality multi-modality data is costly and a non-trivial effort. The high-quality sensors are expensive, and the calibration process demands intensive human efforts. Additionally, real-world driving data suffers from the severe long-tailed distribution problem, making it an obstacle for corner case collection such as in extreme weather conditions. This raises a natural question: *Can we use a controllable way to synthesize aligned multi-modality data?*

Given the success in other fields (Rombach et al., 2022; Blattmann et al., 2023; Liu et al., 2023a), generative models offer a promising solution. Current research focuses either on synthesizing point clouds (Zyrianov et al., 2022; Hu et al., 2024a; Ran et al., 2024) or multi-view images (Gao et al., 2023; Wang et al., 2023; Wen et al., 2024), with limited attention paid to generating multi-modality data. A simple combination of these single-modality algorithms results in serious cross-modality mismatches in the synthetic scenes (Fig. 1a). Such inconsistencies create ambiguities and even contradictory inputs or supervision signals, thus hindering the performance of downstream tasks.

*Cross-modality consistency* serves as the key desiderata of multi-modality data generation. However, there are several challenges in the generation of consistent LiDAR and camera data. Firstly, synthetic point clouds and multi-view images must be spatially aligned in all the local regions since they describe the same driving scene, *i.e.* the shapes and layouts of both foregrounds and backgrounds must be matched. Secondly, unlike those 2D pixel-level tasks (Zhang et al., 2023), point clouds and multi-view images have distinct geometrical spaces and data formats. Multi-view images are represented by RGB values in camera perspective views, while point clouds are XYZ coordinates in the 3D space. Thirdly, the 3D spatial information is ambiguous during generation for both point clouds and multi-view images without reliable point location or pixel depth in the denoising process.

In this paper, we fill in this gap by proposing X-DRIVE, a novel framework for the joint generation of LiDAR and camera data, as demonstrated in Fig. 1b. We design a dual-branch architecture with two latent diffusion models separately dedicated to the synthesis of point clouds and multi-view images, while a key cross-modality condition module enhances *cross-modality consistency* between them. To model the joint distribution and ensure local spatial alignment, we perform the cross-modality conditions locally with only corresponding regions from the other modality considered. An explicit transform bridges the two different geometrical spaces, converting the cross-modality condition to match each other's noisy latent space. To handle the positional ambiguity, we resort to a 3D-aware design based on epipolar lines on range images and multi-view images, allowing the cross-attention module to adaptively determine the cross-modality correspondence without explicit 3D positions. In consequence, X-DRIVE is able to exploit existing single-modality data as conditions to seamlessly generate data in the other modality. Additionally, X-DRIVE enhances controllability by introducing 3D bounding boxes for geometrical layout control and text prompts for attribute control (*e.g.* weather and lighting), enabling more flexible, fine-grained, and precise control over both modalities.

Extensive experiments demonstrate the great ability of X-DRIVE in generating realistic multi-modality sensor data. It notably outperforms previous specialized single-modality algorithms in the quality of both synthetic point clouds and multi-view images. More importantly, for the first time, it demonstrates reliable cross-modality consistency in synthetic scenes with comprehensive and flexible conditions. Our contributions are summarized as follows.

- We introduce X-DRIVE, a dual-branch multi-modality latent diffusion framework that, for the first time, enables controllable and reliable synthesis of aligned LiDAR and multi-view camera data.
- Our cross-modality epipolar condition module bridges the geometrical gap under spatial ambiguity between point clouds and multi-view images, significantly enhancing modality consistency.
- Extensive experimental results demonstrate the effectiveness of X-DRIVE, with notable MMD (for point cloud generation) and FID (for multi-view generation) improvements, establishing a new state-of-the-art for remarkable cross-modality consistency.

## 2 RELATED WORKS

**Conditional generation with diffusion models.** Diffusion models (Ho et al., 2020; Dhariwal & Nichol, 2021; Podell et al., 2023; Peebles & Xie, 2023) exhibit remarkable ability in generating diverse images by learning a progressive denoising process. They yield state-of-the-art results in various tasks such as text-to-image generation (Rombach et al., 2022; Nichol et al., 2021), text-to-video generation (Singer et al., 2022; Yu et al., 2022), and instructional image editing (Brooks et al., 2023; Meng et al., 2022). Beyond text condition, several work emerges with the competence in managing additional forms of control signals. Zhang et al. (2023) integrates spatial conditions to a pretrained text-to-image diffusion model via efficient finetuning. Li et al. (2023); Zheng et al. (2023) condition the image synthesis on the fine-grained geometrical annotations to facilitate downstream tasks like 2D object detection. Liu et al. (2023a); Sargent et al. (2024); Liu et al. (2023b); Yang et al. (2024b) introduce 3D-aware diffusion models for novel view synthesis based on single-view

image inputs, while Shi et al. (2023) explores text-conditioned multi-view image generation. Unlike these prior work for the image domain, our framework simultaneously generates point clouds and multi-view images conditioned on text prompts and 3D bounding boxes.

**Cross-modality data generation.** Beyond image domain, generative models are extended to the synthesis of multi-modality data. Hu et al. (2024b); Shao et al. (2024) leverage video diffusion models to estimate sequential depths. Bai et al. (2024) formulates various vision tasks as next token prediction by extracting diverse visual information as unified visual tokens. The cross-modality synthesis between videos and audios also draws great attention. Efforts are devoted to video-to-audio (Zhu et al., 2022), audio-to-video (Chatterjee & Cherian, 2020), bi-directional (Chen et al., 2017; Hao et al., 2018), and joint multi-modality (Ruan et al., 2023) generation. However, in all the cases mentioned above, there exists clear cross-modality alignments, such as pixel-to-pixel spatial correspondence for depth estimation and frame-to-frame temporal alignment between audios and videos. In contrast, point clouds and multi-view images are in different geometrical spaces without clear one-to-one local correlation between noisy multi-modality samples.

**Generation of vehicle sensor data.** Panoramic driving environment is usually perceived by the multi-view cameras and LiDARs on the ego-vehicle. Alternative to the laborious collection and annotation process, many works resort to the fast-growing generative models for the synthesis of either cameras (Swerdlow et al., 2024; Yang et al., 2023; Gao et al., 2023; Lu et al., 2024; Singh et al., 2024) or LiDAR (Zyrianov et al., 2022; Hu et al., 2024a; Zyrianov et al., 2024) sensor data. For multi-view images (Yang et al., 2023; Wang et al., 2023; Gao et al., 2023) or videos (Wen et al., 2024; Lu et al., 2023), previous work leverages pretrained image diffusion models (Rombach et al., 2022) by incorporating extra inter-view modules to ensure the multi-view consistency. For point clouds, Zyrianov et al. (2022); Ran et al. (2024); Hu et al. (2024a) adapt latent diffusion models for the generation of range images (Fan et al., 2021; Li et al., 2016) since this range-view representation shares a similar format with RGB images. However, all above methods concentrate on the single-modality data, either LiDAR or camera sensors. Despite various control signals, there exists no guarantee for the cross-modality consistency between point clouds and multi-view images generated by independent single-modality models. To this end, we propose to synthesize consistent multi-modality data in a joint manner.

## 3 PRELIMINARY

**Latent Diffusion Models.** Diffusion models (Ho et al., 2020; Sohl-Dickstein et al., 2015) learn to approximate a data distribution $p(\mathbf{x}_0)$ by iteratively denoising a random Gaussian noise for $T$ steps. Typically, diffusion models construct a diffused input $\mathbf{x}_t$ through a forward process, which gradually adds Gaussian noise to the data according to a variance schedule $\beta_1, \ldots, \beta_T$.

$$q(\mathbf{x}_{1:T}|\mathbf{x}_0) = \prod_{t=1}^{T} q(\mathbf{x}_t|\mathbf{x}_{t-1}), \quad q(\mathbf{x}_t|\mathbf{x}_{t-1}) = \mathcal{N}(\mathbf{x}_t; \sqrt{1-\beta_t}\mathbf{x}_{t-1}, \beta_t\mathbf{I}). \tag{1}$$

Then, the reverse process learns to recover the original inputs by fitting $p_\theta(\mathbf{x}_{t-1}|\mathbf{x}_t)$.

$$p_\theta(\mathbf{x}_{0:T}) = p(\mathbf{x}_T) \prod_{t=1}^{T} p_\theta(\mathbf{x}_{t-1}|\mathbf{x}_t), \quad p_\theta(\mathbf{x}_{t-1}|\mathbf{x}_t) = \mathcal{N}(\mathbf{x}_{t-1}; \mu_\theta(\mathbf{x}_t, t), \Sigma_\theta(\mathbf{x}_t, t)). \tag{2}$$

Latent diffusion models (Rombach et al., 2022) perform the diffusion process on the latent space instead of the input space to handle the high dimensional data. Specifically, it maps input $\mathbf{x}$ with an encoder $\mathcal{E}$ into the latent space as $\mathbf{z} = \mathcal{E}(\mathbf{x})$. The latent code $\mathbf{z}$ can be reconstructed to the input as $\hat{\mathbf{x}} = \mathcal{D}(\mathbf{z})$ through a decoder $\mathcal{D}$. The forward and reverse processes of latent diffusion models are similar with the original diffusion models by substituting $\mathbf{x}$ with latent $\mathbf{z}$ in Eq. 1 and Eq. 2.

**Range Image Representation.** Compatible with the sampling process of LiDAR sensor, range image $\mathcal{R} \in \mathbb{R}^{H_r \times W_r \times 2}$ is a dense rectangular representation for LiDAR point clouds, where $H_r$ and $W_r$ are the numbers of rows and columns with one channel representing the range and the other representing intensity. The rows reflect the laser beams while the columns indicate the yaw angles. For each point with Cartesian coordinate $(x, y, z)$, it can be transformed to the spherical coordinates $(r, \theta, \phi)$ through the following projection.

$$r = \sqrt{x^2 + y^2 + z^2}, \quad \theta = \arctan(y, x), \quad \phi = \arctan(z, \sqrt{x^2 + y^2}) \tag{3}$$

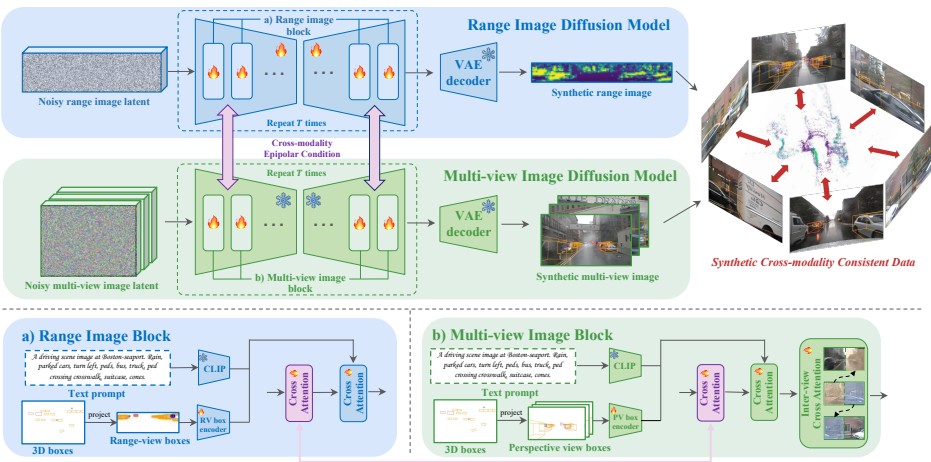

Figure 2: Overview of our proposed X-DRIVE framework. We design a dual-branch diffusion model architecture to generate multi-modality data. Cross-modality epipolar condition modules (Fig. 3) are inserted between branches to enhance the cross-modality consistency.

where $r$ is the range, $\theta$ is the inclination, and $\phi$ is the azimuth. Inversely, the 3D point clouds are recovered from range images by $x = r\cos(\phi)\cos(\theta), y = r\cos(\phi)\sin(\theta), z = r\sin(\phi)$.

The range image is produced by quantizing the $\theta$ and $\phi$ with factors $s_\theta$ and $s_\phi$. For each point with spherical coordinate $(r_i, \theta_i, \phi_i)$, we have $\mathcal{R}(\lfloor \theta_i/s_\theta \rfloor, \lfloor \phi_i/s_\phi \rfloor) = (r_i, i_i)$. The range $r_i$ and intensity $i_i$ are normalized to $(0, 1)$.

## 4 METHODOLOGY

In this section, we present our novel framework, X-DRIVE, for joint generation of point clouds and multi-view images. The overall architecture is depicted in Fig. 2. We extend the vanilla diffusion models to model the joint distribution of multi-modality data in Sec. 4.1. Practically, it is achieved by our proposed framework with dual-branch diffusion models in Sec. 4.2. For the cross-modality consistency, we propose a cross-modality epipolar condition module in Sec. 4.3 . Our method can also be tailored for cross-modality conditional generation in a zero-shot manner as in Sec. 4.4.

### 4.1 JOINT MULTI-MODALITY GENERATION

Beyond single-modality diffusion models in Sec. 3, we give a formulation to our proposed X-DRIVE for multi-modality data generation. It aims to approximate a joint distribution for paired data $(\mathbf{r}_0, \mathcal{C}_0)$ that describes a specific driving scene, where $\mathbf{r}_0$ is the range image representation of point clouds and $\mathcal{C}_0 = \{\mathbf{x}_0^v\}_{v=1}^V$ denotes multi-view images from $V$ cameras at different perspective views.

Given their similar rectangular latent formats, we set a shared noise schedule $\beta_1, \ldots, \beta_T$ for range images and multi-view images. The forward processes of either range image or multi-view images only depend on its own modality, equivalent with independent forward processes for each modality.

$$q(\mathbf{r}_t|\mathbf{r}_{t-1}) = \mathcal{N}\left(\mathbf{r}_t; \sqrt{1-\beta_t}\mathbf{r}_{t-1}, \beta_t\mathbf{I}\right), \quad q(\mathcal{C}_t|\mathcal{C}_{t-1}) = \mathcal{N}\left(\{\mathbf{x}_t^v\}_v; \{\sqrt{1-\beta_t}\mathbf{x}_{t-1}^v\}_v, \{\beta_t\mathbf{I}\}_v\right). \quad (4)$$

In contrast, the reverse process exploits models $\epsilon_{\theta_r}, \epsilon_{\theta_c}$ to predict the noise $\epsilon_r$ and $\epsilon_c$ added to the range image and multi-view images separately. Given the requirements for cross-modality consistency, the noise prediction for each modality should take each other into consideration, so the reverse process for either range image or multi-view images would depend on both modalities.

$$
\begin{aligned}
p_{\theta_r}(\mathbf{r}_{t-1}|\mathbf{r}_t, \mathcal{C}_t) &= \mathcal{N}(\mathbf{r}_{t-1}; \mu_{\theta_r}(\mathbf{r}_t, \mathcal{C}_t, t), \Sigma_{\theta_r}(\mathbf{r}_t, \mathcal{C}_t, t)), \\
p_{\theta_c}(\mathcal{C}_{t-1}|\mathcal{C}_t, \mathbf{r}_t) &= \mathcal{N}(\mathcal{C}_{t-1}; \mu_{\theta_c}(\mathcal{C}_t, \mathbf{r}_t, t), \Sigma_{\theta_c}(\mathcal{C}_t, \mathbf{r}_t, t)).
\end{aligned}
\quad (5)
$$

Intuitively, we can implement two diffusion models as $\epsilon_{\theta_r}$ and $\epsilon_{\theta_c}$ separately for range images and multi-view images, equipped with encoders $c_{CR}$ and $c_{RC}$ for the other modality as extra conditions.

$$\hat{\boldsymbol{\epsilon}}_r = \epsilon_{\theta_r}(\mathbf{r}_t, c_{CR}(\mathcal{C}_t, t), t), \quad \hat{\boldsymbol{\epsilon}}_c = \epsilon_{\theta_c}(\mathcal{C}_t, c_{RC}(\mathbf{r}_t, t), t). \quad (6)$$

Instead of training separate condition encoders, we can directly treat diffusion models themselves as strong encoders for each modality. Thus, $\epsilon_{\theta_c}$ and $\epsilon_{\theta_r}$ are adapted as the condition encoders $\epsilon'_{\theta_c}(\mathcal{C}_t, t)$ and $\epsilon'_{\theta_r}(\mathbf{r}_t, t)$ correspondingly in each block. Compared to text-to-image condition, a notable characteristic of LiDAR-camera mutual condition lies in the local spatial correspondence. For example, a car captured by LiDAR should share the consistent shape and location with the same car captured by cameras but has little direct relationship with other parts in the images. Due to their different geometrical space, we should rely on a spatial transformation $\mathcal{T}(\cdot)$ to explicitly align the geometrical space for cross-modality correspondence. Concretely, $\mathcal{T}_{CR}$ transforms the multi-view image conditions to the range image space, while $\mathcal{T}_{RC}$ transforms the range image conditions to the multi-view image space. As a result, the condition encoders $c_{CR}, c_{RC}$ can be formulated as follows.

$$c_{CR}(\mathcal{C}_t, t) = \mathcal{T}_{CR}(\epsilon'_{\theta_c}(\mathcal{C}_t, t)), \quad c_{RC}(\mathbf{r}_t, t) = \mathcal{T}_{RC}(\epsilon'_{\theta_r}(\mathbf{r}_t, t)). \tag{7}$$

In this case, we rewrite the denoising models in Eq. 6 as follows.

$$\hat{\boldsymbol{\epsilon}}_r = \epsilon_{\theta_r}(\mathbf{r}_t, \mathcal{T}_{CR}(\epsilon'_{\theta_c}(\mathcal{C}_t, t)), t), \quad \hat{\boldsymbol{\epsilon}}_c = \epsilon_{\theta_c}(\mathcal{C}_t, \mathcal{T}_{RC}(\epsilon'_{\theta_r}(\mathbf{r}_t, t)), t). \tag{8}$$

They can be trained with a joint multi-modality objective function $\mathcal{L}_{DM-M}$.

$$\mathcal{L}_{DM-M} = \mathcal{L}_{DM-R} + \mathcal{L}_{DM-C}, \quad \mathcal{L}_{DM-R} = \mathbb{E}_{r,t,\epsilon_r}||\epsilon_r - \hat{\epsilon}_r||_2^2, \quad \mathcal{L}_{DM-C} = \mathbb{E}_{\mathcal{C},t,\epsilon_c}||\epsilon_c - \hat{\epsilon}_c||_2^2. \tag{9}$$

## 4.2 DUAL-BRANCH JOINT GENERATION FRAMEWORK

Our framework is developed following Eq. 8, which consists of two denoising models, *i.e.*, $\epsilon_{\theta_r}(\cdot)$ and $\epsilon_{\theta_c}(\cdot)$ respectively tailored for range images (Sec. 4.2.1) and multi-view images (Sec. 4.2.2).

### 4.2.1 DIFFUSION MODEL FOR RANGE IMAGES

**Architecture.** We adapt existing latent diffusion model (LDM) (Rombach et al., 2022) to learning the distribution of range image $\mathbf{r_0} \in \mathbb{R}^{H_r \times W_r \times 2}$ given its similar representation format with RGB images. In the first stage, the range image is compressed into latent feature $\mathbf{z}_0^r \in \mathbb{R}^{h_r \times w_r \times c_r}$ by a downsampling factor $f_r = H_r/h_r = W_r/w_r$. In the second stage, we train an LDM for range images from scratch to learn the latent distribution of $\mathbf{z}_0^r$. Range image reflects the panoramic view of the ego-vehicle, so its left-most and right-most sides are connected. Considering this constraint, we follow LiDARGen (Zyrianov et al., 2022) and RangeLDM (Hu et al., 2024a) to replace all the convolutions in both VAE and LDM with horizontally circular convolutions (Schubert et al., 2019) where the left and right sides of the range image are treated as neighbors.

In the first stage, we follow the standard protocol to train the VAE including both encoder $\mathcal{E}_r$ and decoder $\mathcal{D}_r$ by maximizing the ELBO (Rezende et al., 2014). In addition, we attach an adversarial discriminator (Isola et al., 2017; Rombach et al., 2022) to mitigate the blurriness brought by the reconstruction loss. In the second stage, the LDM conditions the synthesis of range images on the cross-modality information from the multi-view image branch (elaborated in Sec. 4.3) as well as the text prompt and 3D boxes. The model is trained with loss function in Eq. 9.

**Range-view bounding box condition.** Given a 3D bounding box $(\mathbf{b}_i, c_i)$, $\mathbf{b}_i = \{(x_j, y_j, z_j)\}_{j=1}^8 \in \mathbb{R}^{8 \times 3}$ represents the 3D coordinates of box corners and $c_i$ denotes its semantic category. We project each box corner to the range view using Eq. 3 as $\mathbf{b}_i^r = \{(r_j, \theta_j, \phi_j)\}_{j=1}^8$. These range-view corner coordinates are passed through a Fourier embedder. Then, the concatenated position embeddings of eight corners are encoded by $\mathrm{MLP}_p^r$ as $\mathbf{h}_{p,i}^r$. For category label $c_i$, we follow (Isola et al., 2017; Gao et al., 2023) to pool the CLIP embedding (Radford et al., 2021) of the category name as $\mathbf{h}_{l,i}^r$. The box and label embeddings are combined and encoded by another $\mathrm{MLP}_b^r$ as a hidden vector $\mathbf{h}_{b,i}^r$.

$$\mathbf{h}_{b,i}^r = \mathrm{MLP}_b^r([\mathbf{h}_{l,i}^r, \mathbf{h}_{p,i}^r]), \quad \mathbf{h}_{l,i}^r = \mathrm{Avg}(\mathrm{CLIP}(c_i)), \quad \mathbf{h}_{p,i}^r = \mathrm{MLP}_p^r(\mathrm{Concat}(\mathrm{Fourier}(\mathbf{b}_i^r))) \tag{10}$$

We concatenate box hidden vectors $\{\mathbf{h}_{b,i}^r\}_i$ from this range-view box encoder with text prompt hidden vectors, which guide the synthesis the latent diffusion model via a cross-attention module.

### 4.2.2 DIFFUSION MODEL FOR MULTI-VIEW IMAGES

**Architecture.** To model the distribution of multi-view images $\mathcal{C}_0 = \{\mathbf{x}_0^v\}_{v=1}^V$, $\mathbf{x}_0^v \in \mathbb{R}^{H_c \times W_c \times 3}$, we tailor the latent diffusion model by introducing extra bounding box and inter-view conditions apart from the cross-modality signals from range images (details in Sec. 4.3). Following standard

Figure 3: Cross-modality epipolar condition module. We perform mutual conditions locally between LiDAR and camera modalities based on epipolar lines on multi-view image and range image latents.

LDM (Rombach et al., 2022), each image $\mathbf{x}_0^v$ is first compressed by a pretrained VAE to the latent space as $\mathbf{z}_0^{c,v}$. In the training process, we sample the Gaussian noise for each view independently and the denoising model is trained with the objective function in Eq. 9.

**Multi-view bounding box condition.** The 3D box condition module is similar with Sec. 4.2.1, but the box positions are transformed to the perspective view of each camera. For each 3D bounding box $(\mathbf{b}_i, c_i)$, we project its 3D corner coordinates $\mathbf{b}_i = \{(x_j, y_j, z_j)\}_{j=1}^8$ to corresponding camera perspective view as $\mathbf{b}_i^c = \{(u_j, v_j, d_j)\}_{j=1}^8$, where $u_j, v_j$ are the corner location in the pixel coordinate and $d_j$ is the depth. For each camera view $v$, its synthesis is only conditioned on the 3D bounding boxes with at least one corner projected into the range of its perspective view image. Then, we formulate this perspective-view box encoder similar with Eq. 10.

$$\mathbf{h}_{b,i}^c = \text{MLP}_b^c([\mathbf{h}_{l,i}^c, \mathbf{h}_{p,i}^c]), \quad \mathbf{h}_{l,i}^c = \text{Avg}(\text{CLIP}(c_i)), \quad \mathbf{h}_{p,i}^c = \text{MLP}_p^c(\text{Concat}(\text{Fourier}(\mathbf{b}_i^c))) \quad (11)$$

For each camera view, the box embeddings $\{\mathbf{h}_{b,i}^c\}_i$ are concatenated with the text prompt hidden vectors, which together guide the image generation through a cross-attention module.

**Inter-view condition.** It is critical to enhance the consistency across different views in the generation of multi-view images $\mathcal{C}_0 = \{\mathbf{x}_0^v\}_{v=1}^V$, so we inject inter-view cross-attention module to condition the generation of each camera view $v$ on its left and right adjacent views $vl$ and $vr$. Given the small overlapping between field-of-views of adjacent cameras, we split each image latent $\mathbf{z}_{in}^{c,v}$ into left and right halves. For each view $v$, its left half $\mathbf{z}_{in,l}^{c,v}$ attends to the right half $\mathbf{z}_{in,r}^{c,vl}$ of its left neighbor view $vl$, while its right half $\mathbf{z}_{in,r}^{c,v}$ depends on the left half $\mathbf{z}_{in,l}^{c,vr}$ of its right neighbor $vr$.

$$\mathbf{z}_{out}^{c,v} = \mathbf{z}_{in}^{c,v} + \tanh(\alpha_v) \cdot \text{Concat}_{width}\left(\left[\text{CrossAttn}(\mathbf{z}_{in,l}^{c,v}, \mathbf{z}_{in,r}^{c,vl}); \text{CrossAttn}(\mathbf{z}_{in,r}^{c,v}, \mathbf{z}_{in,l}^{c,vr})\right]\right) \quad (12)$$

where $\alpha_v$ is a zero-initialization gate (Zhang et al., 2023) for stable optimization. Compared to full cross-attention in MagicDrive (Gao et al., 2023), our split strategy significantly reduces the per-scene GPU memory cost of inter-view condition from 11GB to 3GB with better multi-view consistency.

## 4.3 CROSS-MODALITY EPIPOLAR CONDITION MODULE

The key to multi-modality data synthesis is to boost the cross-modality consistency, which potentially relies on cross-modality conditions. Ideally, there is an explicit point-to-pixel correspondence between point clouds and multi-view images through camera projection. However, in the denoising process, we are not aware of either the range values of range images or the depths of the multi-view images. Instead, we propose to warp the local features from range images and multi-view images based on epipolar lines through spatial transforms $\mathcal{T}_{RC}$ and $\mathcal{T}_{CR}$ (Eq. 7) to provide control signals for the synthesis of the other modality, illustrated in Fig. 3.

**Camera-to-LiDAR condition.** For each position $(\phi, \theta)$ on the range image latent $\mathbf{z}_{in}^r$, we sample $R$ points along the range axis following the linear-increasing discretization (LID) (Reading et al., 2021; Liu et al., 2022) with range value $r_k, k = 1, 2, \ldots, R$. We transform these points $\{(\phi, \theta, r_k)\}_{k=1}^R$ to their 3D coordinates $\{(x_k, y_k, z_k)\}_{k=1}^R$.

$$x_k = r_k \cdot \cos(\phi) \cdot \sin(\theta), \quad y_k = r_k \cdot \cos(\phi) \cdot \cos(\theta), \quad z_k = r_k \cdot \sin(\phi). \quad (13)$$

They can be projected to the camera perspective view $v$ with the camera parameters $(\mathbf{R}_v, \mathbf{t}_v, \mathbf{K}_v)$ as

$$d_k \begin{bmatrix} u_k & v_k & 1 \end{bmatrix}^T = \mathbf{K}_v\left(\mathbf{R}_v \begin{bmatrix} x_k & y_k & z_k \end{bmatrix}^T + \mathbf{t}_v\right), \quad k = 1, 2, \ldots, R \quad (14)$$

where $(u_k, v_k), k = 1, 2, \ldots, R$ are the pixel coordinates for the $R$ points along the LiDAR ray. They form the epipolar line on the camera image $\mathbf{x}^v$ corresponding to position $(\phi, \theta)$ of the LiDAR

range image. We bilinearly sample the corresponding local features from the image latent $\mathbf{z}_{in}^{c,v}$ of view $v$ at location $(u_k, v_k)$. If a 3D point $(x_k, y_k, z_k)$ is projected into more than one camera view, we would simply adopt the average local features. Extra Fourier embedding of $r_k$ is added to each sampled feature as an indicator for the range value along the ray. As a result, the entire spatial transform $\mathcal{T}_{CR}$ makes the camera image features spatially aligned to the range image features.

$$\mathcal{T}_{CR}(\mathbf{z}_{in}^{c,v})_{(\phi,\theta)} = \{\text{Bilinear-Sample}(\mathbf{z}_{in}^{c,v}; (u_k, v_k)) + \text{MLP}_r\left(\text{Fourier}(r_k)\right)\}_{k=1}^{R}. \tag{15}$$

We apply a cross-attention module to condition the range image feature $\mathbf{z}_{in,(\phi,\theta)}^{r}$ at coordinate $(\phi, \theta)$ on the corresponding transformed local image features $\mathcal{T}_{CR}(\mathbf{z}_{in}^{c})_{(\phi,\theta)}$ on the epipolar line.

$$\mathbf{z}_{out,(\phi,\theta)}^{r} = \mathbf{z}_{in,(\phi,\theta)}^{r} + \tanh(\alpha_{cr}) \cdot \text{CrossAttn}\left(\mathbf{z}_{in,(\phi,\theta)}^{r}, \mathcal{T}_{CR}(\mathbf{z}_{in}^{c,v})_{(\phi,\theta)}\right) \tag{16}$$

where $\alpha_{cr}$ is a zero-initialization gate for the camera-to-LiDAR condition. The model learns the adaptive local correspondence between range image and multi-view images under range ambiguity.

**LiDAR-to-camera condition.** Inversely, we condition the synthesis of multi-view images on the LiDAR range images using a similar module. For each position $(u, v)$ on the camera image latent $\mathbf{z}_{in}^{c,v}$ from view $v$, $D$ points are sampled at different depths following the linear-increasing discretization with depth value $d_k, k = 1, 2, \ldots, D$. Each point $(u, v, d_k)$ in the pixel coordinate is transformed to 3D coordinate $(x_k, y_k, z_k)$ via the corresponding camera parameters $(\mathbf{R}_v, \mathbf{t}_v, \mathbf{K}_v)$.

$$\begin{bmatrix} x_k & y_k & z_k \end{bmatrix}^T = \mathbf{R}_v^T \left( \mathbf{K}_v^{-1} \begin{bmatrix} u_k \cdot d_k & v_k \cdot d_k & d_k \end{bmatrix}^T - \mathbf{t}_v \right), \quad k = 1, 2, \ldots, D \tag{17}$$

The 3D point coordinate is then projected to the LiDAR range view using Eq. 3 as $(\phi_k, \theta_k, r_k)$. The $D$ points $(\phi_k, \theta_k), k = 1, 2, \ldots, D$ form the epipolar line on the LiDAR range image corresponding to the pixel coordinate $(u, v)$ of camera view $v$. The local range image features are extracted at each coordinate $(\phi_k, \theta_k)$ from range image latent $\mathbf{z}_{in}^{r}$ through bilinear sampling with additional Fourier embedding for the depth value $d_k, k = 1, 2, \ldots, D$ attached. We can write the spatial transform $\mathcal{T}_{RC}$ aligning the range view features to multi-view image features as follows.

$$\mathcal{T}_{RC}(\mathbf{z}_{in}^{r})_{(u,v)} = \{\text{Bilinear-Sample}(\mathbf{z}_{in}^{r}; (\theta_k, \phi_k)) + \text{MLP}_d\left(\text{Fourier}(d_k)\right)\}_{k=1}^{D}. \tag{18}$$

Similar with camera-to-LiDAR condition, a cross-attention module conditions the local camera image feature $\mathbf{z}_{out,(u,v)}^{c,v}$ at coordinate $(u, v)$ on spatially aligned range image features $\mathcal{T}_{RC}(\mathbf{z}_{in}^{c,v})_{(u,v)}$.

$$\mathbf{z}_{out,(u,v)}^{c,v} = \mathbf{z}_{in,(u,v)}^{c,v} + \tanh(\alpha_{rc}) \cdot \text{CrossAttn}\left(\mathbf{z}_{in,(u,v)}^{c,v}, \mathcal{T}_{RC}(\mathbf{z}_{in}^{r})_{(u,v)}\right) \tag{19}$$

where $\alpha_{rc}$ is a zero-initialization gate. The cross-attention module adaptively learns the local reliance of camera multi-view images on LiDAR range images without explicit depth cues.

## 4.4 CROSS-MODALITY CONDITIONAL GENERATION

Besides multi-modality joint generation, X-DRIVE can work as a camera-to-LiDAR or LiDAR-to-camera conditional generation model in a zero-shot manner without targeted training. Given either one of ground-truth LiDAR range image $\mathbf{r}_0$ or multi-view images $\mathcal{C}_0$, we change Eq. 8 separately for camera-to-LiDAR or LiDAR-to-camera generation to synthesize the other modality data.

$$\hat{\boldsymbol{\epsilon}}_r = \epsilon_{\theta_r}(\mathbf{r}_t, \mathcal{T}_{CR}(\epsilon_{\theta_c}'(\mathcal{C}_0, 0)), t), \quad \hat{\boldsymbol{\epsilon}}_c = \epsilon_{\theta_c}(\mathcal{C}_t, \mathcal{T}_{RC}(\epsilon_{\theta_r}'(\mathbf{r}_0, 0)), t). \tag{20}$$

In this case, the trained diffusion model for the input modality serves as a strong encoder for the condition to enhance the adherence of generation to the input LiDAR or camera conditions.

## 5 EXPERIMENTS

### 5.1 EXPERIMENTAL SETUPS

**Dataset.** We evaluate our method using nuScenes dataset (Caesar et al., 2020). It provides point clouds from a 32-beam LiDAR and multi-view images from 6 cameras. We follow the official setting to employ 700 driving scenes for training and 150 scenes for validation. The generation of X-DRIVE is conditioned on bounding boxes from 10 semantic classes as well as scene descriptions.

**Evaluation Metrics.** For joint multi-modality data synthesis, we evaluate both single-modality data realism and cross-modality consistency. For image modality, we report Fréchet Inception Distance (FID) for image synthesis realism. We also utilize View Consistency Score (VSC) (Swerdlow et al., 2024) to evaluate multi-view consistency, which reflects the keypoints correspondence on overlapping regions between adjacent camera views (Fig. 5), and we normalize the VSC of real images to 1.0 for clarity. For point clouds modality, we measure the distribution gap between synthetic and real point clouds with Maximum Mean Discrepancy (MMD) and Jensen-Shannon divergence (JSD) following RangeLDM (Hu et al., 2024a). For cross-modality consistency, we propose a novel metric called Depth Alignment Score (DAS). We project the synthetic point clouds to multi-view images as sparse depth values and also estimate the image depth using DepthAnythingV2 (Yang et al., 2024d). The mean absolute error between the projected and estimated disparities is used as DAS metric.

**Baselines.** To our best knowledge, we are the first to work on the joint generation of point clouds and multi-view images. For single-modality generation, we compare with state-of-the-art generation algorithms for point clouds, *i.e.* LiDAR VAE (Caccia et al., 2019), LiDARGen (Zyrianov et al., 2022), RangeLDM (Hu et al., 2024a), and multi-view images, *i.e.* BEVGen (Swerdlow et al., 2024), BEV-Control (Yang et al., 2023), MagicDrive (Gao et al., 2023), with released code or quantitative results on nuScenes dataset. Furthermore, we combine MagicDrive (Gao et al., 2023) and RangeLDM (Hu et al., 2024a) respectively for images and point clouds as a multi-modality baseline.

**Training Setup.** Our X-DRIVE has a dual-branch architecture. We utilize the Stable-Diffusion pretrained weight to initialize the multi-view image branch with other newly added parameters randomly initialized. We follow a three-stage pipeline for the training. In the first stage, the VAE is trained from scratch for range images. In the second stage, we train the latent diffusion model in range image branch without cross-modality conditions using the frozen VAE. In the last stage, the entire multi-modality model is trained in an end-to-end manner with only the Stable-Diffusion parts frozen. The first two stages can also be replaced with pretrained point clouds diffusion model. We follow MagicDrive (Gao et al., 2023) and RangeLDM (Hu et al., 2024a) to synthesize $224 \times 400$ multi-view camera images and $32 \times 1024$ point cloud range images.

## 5.2 QUANTITATIVE RESULTS

We report the quantitative results for both multi-modality generation as well as LiDAR-to-camera or camera-to-LiDAR cross-modality generation.

**Joint multi-modality generation.** As shown in Tab. 1, compared to specialized single-modality generation methods, X-DRIVE$_{L+C}$, as a multi-modality algorithm, achieves comparable or even better quality in both synthetic point clouds and multi-view images. Our FID metric is outperformed by MagicDrive (Gao et al., 2023), since we do not have map condition which offers strong control signals. Unlike simple combination of single-modality methods, we can generate point clouds and images with cross-modality alignment, reflected by our superior DAS metric, thanks to our proposed cross-modality epipolar condition module.

Table 1: Quantitative comparison with driving data generation algorithms. For each column, the best value is highlighted by **bold**, and the second best is denoted by underline.

(a) Image quality.

| Methods | FID↓ | VSC↑ |
|---|---|---|
| LayoutDiffusion | 29.64 | - |
| WoVoGen | 27.60 | - |
| BEVControl | 25.54 | - |
| BEVGen | 24.85 | 0.439 |
| MagicDrive | 16.20 | 0.633 |
| X-DRIVE$_{L \to C}$ | **16.01** | **0.721** |
| X-DRIVE$_{L+C}$ | 17.37 | 0.675 |

(b) Point clouds quality.

| Methods | MMD↓ | JSD↓ |
|---|---|---|
| LiDAR VAE | $1.1 \times 10^{-3}$ | - |
| LiDARGen | $1.9 \times 10^{-3}$ | 0.160 |
| RangeLDM | $1.9 \times 10^{-4}$ | 0.054 |
| X-DRIVE$_{C \to L}$ | $\mathbf{1.0 \times 10^{-4}}$ | **0.052** |
| X-DRIVE$_{L+C}$ | $\underline{1.2 \times 10^{-4}}$ | **0.052** |

(c) Cross-modality alignment.

| Methods | DAS↓ |
|---|---|
| MagicDrive + RangeLDM | 2.32 |
| X-DRIVE$_{L+C}$ | **1.69** |

**Conditional cross-modality generation.** As mentioned in Sec. 4.4, X-DRIVE can also work as a LiDAR-to-camera or camera-to-LiDAR conditional generation model. In Tab. 1, for single-modality data generation, X-DRIVE can outperform previous baselines for both point clouds and multi-view images, demonstrating the flexibility of our proposed algorithm for cross-modality data synthesis.

**Object-level controllable generation.** X-DRIVE can adhere to the 3D bounding box conditions in the generative process. Since previous LiDAR generation methods do not show this ability (Hu et al., 2024a; Ran et al., 2024), we only compare our methods with multi-view image generation algorithm (Gao et al., 2023). For fair comparison with single-modality method, we employ synthetic multi-view images and real point clouds from nuScenes validation set to run a

Table 2: Object-level control.

| Methods | mAP | NDS |
|---|---|---|
| Oracle | 70.5 | 72.8 |
| MagicDrive | 65.2 | 69.2 |
| X-DRIVE (ours) | **65.4** | **69.6** |

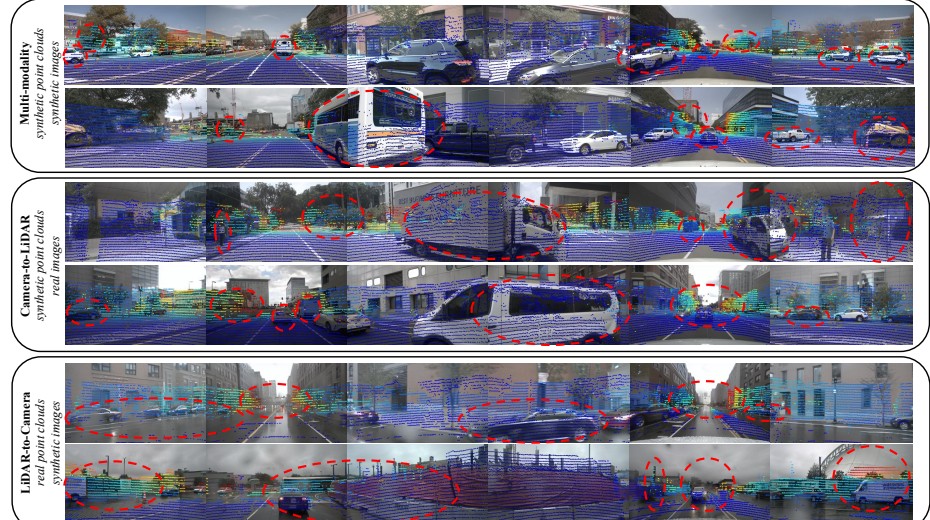

Figure 4: Qualitative results for multi-modality and conditional cross-modality generation. Colors of projected point clouds refer to depths. Well-matched regions are highlighted with red circles.

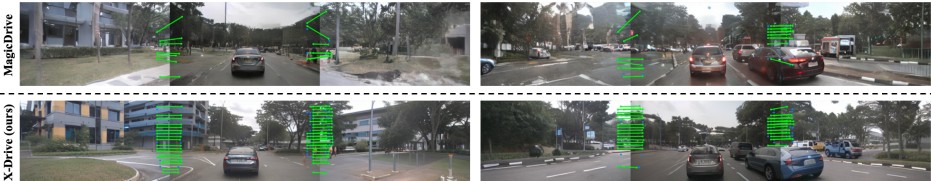

Figure 5: Key points correspondence between adjacent synthetic camera images. Our proposed method (bottom) can bring higher multi-view consistency than the baseline (top).

pretrained SparseFusion (Xie et al., 2023) multi-sensor object detection algorithm. Tab. 2 shows X-DRIVE outperforms MagicDrive (Gao et al., 2023) *w.r.t.* the object-level fidelity.

**Training support for downstream task.** X-DRIVE can produce synthetic multi-modality data to enhance the training for downstream perception models. We train a multi-sensor 3D object detection model using SparseFusion method (Xie et al., 2023) with a combination of $1.4k$

Table 3: Object detection training.

| Training data | mAP | NDS |
|---|---|---|
| real data only | 56.28 | 51.54 |
| real + synthetic data | **57.60** | **52.23** |

real scenes from nuScenes dataset (Caesar et al., 2020) and $1.4k$ synthetic scenes. Tab. 3 shows that the model trained with mixed data can notably outperform the model trained only with real data. This reflects both single-modality quality and cross-modality alignment of our synthetic data.

### 5.3 QUALITATIVE ANALYSIS

**Cross-modality consistency.** We show some multi-modality generation examples to exhibit the cross-modality consistency qualitatively. As in Fig. 4, the projected point clouds overlap with multi-view image contents properly for both foregrounds and backgrounds. We also show the results of conditional camera-to-LiDAR and LiDAR-to-camera generation in Fig. 4, where the synthetic data adheres to the cross-modality LiDAR or camera conditions well.

**Multi-view consistency.** In our multi-modality generation, point clouds provide a natural 3D-aware geometrical guidance for the multi-view images, benefiting the multi-view consistency. In Fig. 5, we visualize the keypoints matching results (Sun et al., 2021) between adjacent synthetic camera images, where better correspondence is witnessed for X-DRIVE than MagicDrive (Gao et al., 2023).

**Scene-level and object-level control.** Given the text and bounding box conditions, X-DRIVE generates diverse results using different control signals, as illustrated in Fig. 6. For scene-level control, we can target for various lighting and weathers through the text prompts. Synthetic images still remain realistic and adhere to the object layouts, and synthetic point clouds become noisy in rainy weather due to the water in the air or on the ground. For object-level control, we can edit the scene by deleting objects or inserting objects at desired locations with specific sizes and semantic categories.

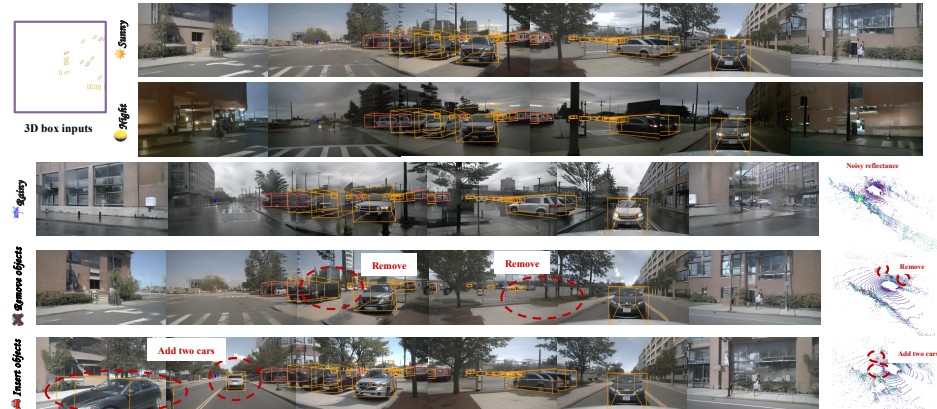

Figure 6: Scene-level and object-level controllable generation by changing the input conditions.

Table 4: Ablations of conditions and training.

| Methods | FID↓ | JSD↓ | DAS↓ |
|---|---|---|---|
| *w/o* 3D box condition | 29.60 | 0.091 | 1.76 |
| *w/o* text prompt condition | 25.76 | 0.070 | 1.73 |
| *w/o* two-stage training | 23.65 | 0.113 | 1.84 |
| full model | **20.17** | **0.070** | **1.67** |

Table 5: Ablations of cross-modality module.

| Methods | FID↓ | JSD↓ | DAS↓ |
|---|---|---|---|
| *w/o* cross-modality | 24.41 | 0.081 | 1.95 |
| *w/o* epipolar line | 25.35 | 0.074 | 2.23 |
| average epipolar condition | 20.70 | 0.098 | 2.23 |
| half sampling points | 22.31 | 0.072 | 1.93 |
| full model | **20.17** | **0.070** | **1.67** |

## 5.4 ABLATION STUDIES

Due to limited resources, we employ shorter training schedule for ablation studies.

**Two-stage training.** X-DRIVE benefits from the two-stage training pipeline that trains the LiDAR range image branch and cross-modality modules in a sequential manner. In Tab. 4, there is a notable performance drop in point clouds quality if we merge the two training stages, since the two-stage training can make the optimization more simple and stable for the LiDAR range image branch.

**Bounding box and text prompt input conditions.** Although X-DRIVE also supports unconditional generation, these input conditions provide multi-level control signals to generate diverse outputs. Tab. 4 shows that text and box conditions improve the quality of synthetic point clouds and images.

**Cross-modality condition module.** The cross-modality condition module is critical for cross-modality consistency. As shown in Tab. 5, removing it significantly hurts the cross-modality alignment although the shared bounding boxes and text prompts can provide some constraints.

**Epipolar-line cross-attention.** Tab. 5 justifies our design of epipolar-line cross-attention module to solve the spatial ambiguity (Eq. 16 and Eq. 19). Alternatively, randomly sampling same number of local features from the other modality feature map without epipolar lines cannot ensure the cross-modality alignment. If we reduce the number of sampling points along the epipolar line, the performance would also be severely hurt. Besides, the cross-attention is also important for adaptively learning the local correspondence. Simple averaging along epipolar lines instead of cross-attention is quite noisy and poses severe damage to the point clouds quality and cross-modality consistency.

## 6 CONCLUSIONS

We present X-DRIVE, a novel framework for the multi-modality generation of aligned LiDAR point clouds and multi-view camera images. By leveraging a dual-branch diffusion model, X-DRIVE captures the joint distribution of multi-modality data, ensuring mutual dependence between modalities. A key feature is the cross-modality condition module, which uses epipolar lines on LiDAR range images and camera views to address spatial ambiguity during the denoising process. Additionally, X-DRIVE supports multi-level control, enabling synthesis based on text prompts and object bounding boxes. Extensive experiments show that X-DRIVE generates high-quality, geometrically and semantically aligned point clouds and images, maintaining fidelity to the input conditions while accurately describing the same driving scene together.

**Acknowledegements** This work was supported in part by Berkeley DeepDrive and NSF Grant. 2235013.

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

In this appendix, we first give an extra ablation study about the classifier-free guidance in Appendix A. Then, we elaborate our implementation details in Appendix B. Afterwards, we discuss the limitation of our framework and future works in Appendix C. Finally, more qualitative results of multi-modality data generation are demonstrated in Appendix D.

## A    ABLATION STUDY OF CLASSIFIER-FREE GUIDANCE SCALES

We employ classifier-free guidance (CFG) during inference for box and text conditions. Illustrated in Fig. 7, it improves the realism of synthetic data. However, higher CFG scale would hurt the image and point cloud quality by increasing the contrast and sharpness.

## B    IMPLEMENTATION DETAILS

In this part, we explain details in the implementation of X-DRIVE including network architecture, training schedule, and evaluation metrics. Our code will be made publicly available upon the acceptance of this paper.

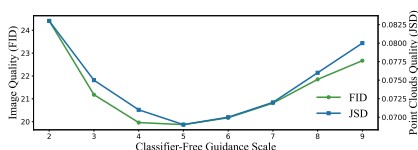

Figure 7: Effect of CFG scales.

**Network architecture.** Given the small height dimension of LiDAR range images and limited data amount, we employ smaller sizes for the VAE and latent diffusion model (LDM) for LiDAR range images. Specifically, the VAE includes three encoder blocks and three decoder blocks with channels $[32, 64, 128]$ separately. There are downsampling modules in the first two encoder blocks. The latent $\mathbf{z}^r$ for range image has a size of $8 \times 256 \times 8$. The range image LDM is composed of three downsampling blocks, one middle block, and three upsampling blocks with attention modules in all the blocks. The channel numbers for three downsampling and upsampling blocks are $[128, 256, 512]$ respectively. The cross-modality epipolar condition modules are inserted between each pair of corresponding blocks in point cloud and multi-view image diffusion model branches. For these seven cross-modality modules, we set the sampling number $R$ and $D$ along range and depth axes as $[24, 24, 12, 6, 12, 24, 24]$ separately with both maximal depth and range as $60m$.

**Training schedule.** The training of X-DRIVE on nuScenes dataset (Caesar et al., 2020) includes three stages. Since there are no publicly available pretrained weights for LiDAR diffusion models, we train the VAE and LDM for LiDAR range image in the first two stages separately. Afterwards, we train the entire multi-modality generation framework together. Before training, we initialize our multi-view image diffusion model branch with pretrained Stable-Diffusion V1.5 weights, while all the other modules except the zero-initialization gates are initialized randomly. In all the stages, our model is trained using NVIDIA RTX A6000 GPUs.

In the first stage, VAE for LiDAR range image is trained using batch size 96 and learning rate 4e-4 for 200 epochs. The discriminator takes effect after 1000 iterations. In the second stage, we train the LiDAR LDM from scratch using batch size 96 and learning rate 1e-4 for 2000 epochs. The model includes the text prompt and 3D range-view bounding box condition modules with drop-rate 0.25 for either condition during training. In the last stage, we train the entire framework in an end-to-end manner with module trainable states illustrated in Fig. 2 using condition drop-rate 0.25 as well. The entire model is trained for 250 epochs with learning rate 8e-5 and batch size 24 in our main experiments. For ablation studies, we reduce the epoch number to 80 for efficiency.

**Evaluation metrics.** In our quantitative evaluation, we apply the FID metric from Magic-Drive (Gao et al., 2023), VSC metric from BEVGen (Swerdlow et al., 2024), MMD and JSD metrics from LiDARGen (Zyrianov et al., 2022). For our proposed DAS metric, we run pretrained DepthAnythingV2 model (Yang et al., 2024d) with ViT-B backbone (Dosovitskiy, 2020) on the synthetic images to estimate the disparity (Yang et al., 2024c). Then, we project the synthetic point clouds to multi-view images to get sparse disparity. We normalize the projected sparse disparity to the same scale as the estimated disparity. The mean absolute error is fetched as our DAS metric.

## C    LIMITATION AND FUTURE WORKS

This paper concentrates on the joint generation of consistent multi-modality data. In this case, due to the limitation of accessible computational resources, we do not include some natural extensions of our X-DRIVE framework. Firstly, our method is limited to the generation of single-frame point clouds and multi-view images. In fact, it is intuitive to combine existing temporal attention modules like Wen et al. (2024) with our X-DRIVE. Secondly, we can also incorporate some extra conditions for data synthesis such as HD maps (Gao et al., 2023) into our framework for more controllable generation. Lastly, due to the distribution of nuScenes training set (Caesar et al., 2020), we cannot synthesize some specific scenarios such as snowy weathers. We will try to extend our framework to deal with above limitations.

Our proposed X-DRIVE is orthogonal to other efforts in improving single-modality generative model performance such as transformer-based image generation framework (Peebles & Xie, 2023), high image resolutions (Xie et al., 2024a), or point clouds representation for LiDAR sensor data. We will combine the cross-modality condition module of X-DRIVE with these advanced techniques to further enhance its performance in multi-modality data generation.

## D    QUALITATIVE RESULTS

In this section, we demonstrate additional examples of our synthetic multi-modality data in Fig. 8. X-DRIVE can generate realistic point clouds and multi-view images with cross-modality consistency based on the input conditions.

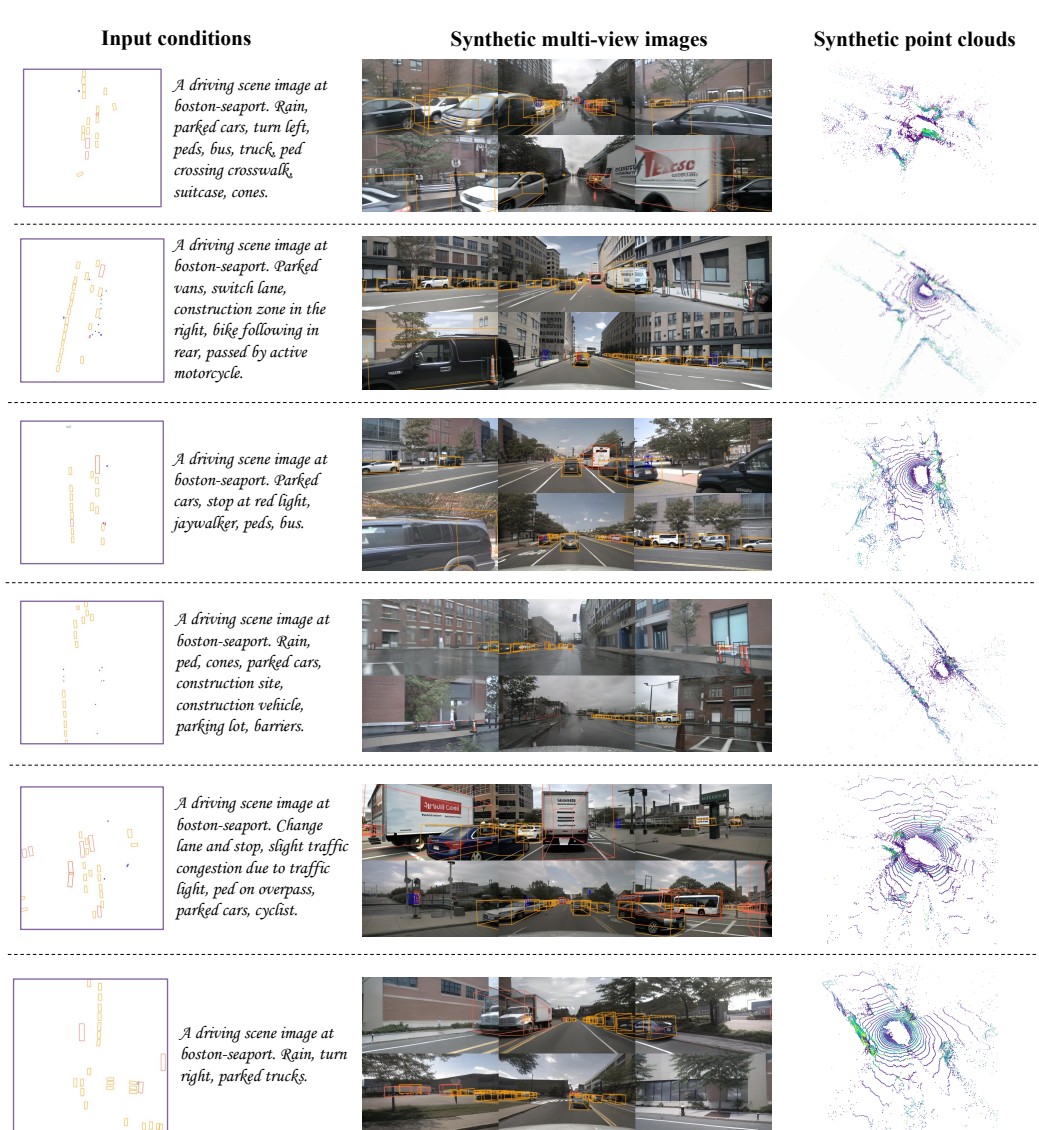

Figure 8: Additional multi-modality data synthesis results using our proposed X-DRIVE.

