# OpenReview forum: "X-Drive: Cross-modality Consistent Multi-Sensor Data Synthesis for Driving Scenarios"
_ICLR.cc/2025/Conference — ICLR 2025 Poster_

### Official Review · Reviewer_SjU7 · 2024-10-27

**Soundness:** 3
**Presentation:** 3
**Contribution:** 3
**Rating:** 6
**Confidence:** 4

**Summary:**

# Review Analysis of X-DRIVE Paper

## 1. Methodology and Key Innovations

### Primary Methods
- Proposed X-DRIVE framework: A dual-branch latent diffusion model architecture for simultaneous generation of LiDAR point clouds and multi-view camera images
- Designed Cross-modality Epipolar Condition Module to enhance consistency between different modalities
- Introduced 3D bounding boxes for geometric layout control and text prompts for attribute control (e.g. weather and lighting)

### Key Innovations
- First work to achieve joint generation of LiDAR point clouds and multi-view camera images with cross-modal consistency
- Novel epipolar-based cross-modal condition module addressing alignment between point clouds and images in different geometric spaces
- Multi-level controllable generation incorporating text, bounding boxes, images and point clouds as conditions

## 2. Innovation Analysis

### Limitations in Innovation
- Weak theoretical foundation for cross-modal consistency, lacking rigorous proof of epipolar-based design effectiveness
- Incomplete solution for "spatial ambiguity", relying solely on epipolar geometry may not fully resolve 3D space ambiguities
- Limited innovation in generation control, mainly applying existing control methods to new scenarios

## 3. Experimental Analysis

### Required Additional Experiments
- Evaluation of generated data utility in downstream tasks (3D object detection, trajectory prediction)
- Need to adopt more advanced transformer architectures like DiT instead of U-Net
- Consider replacing CLIP with T5 for potentially better performance
- Address low resolution issues affecting geometric shapes of distant objects (The current resolution is too low. Objects in the distance have no geometric shapes at all. However, the point cloud may have relatively complete and accurate geometric shapes in the distance. A larger resolution should be used, and because the generated image has slightly less geometric shapes in the distance, the effect is not good, simple and practical super-resolution model. It can only increase the resolution but cannot improve the distant geometry.)
- Investigate video generation capabilities with 3D VAE and 3D attention mechanisms
- Develop quantitative metrics for cross-modal consistency evaluation

## 4. Ablation Study Analysis

### Missing Ablation Studies
- Detailed component-wise analysis of cross-modal epipolar condition module
- Impact assessment of different condition types on generation quality
- Sensitivity analysis of key hyperparameters
- Investigation of different training strategies

## 5. Advanced Experiments (Optional)

### Long-tail Data Generation
- Potential for generating rare/corner cases
- Ability to synthesize challenging scenarios
- Evaluation of model performance on long-tail distributions

## Overall Assessment
While the paper presents novel contributions in cross-modal data generation, it requires strengthening in theoretical foundations, experimental validation, and analysis depth. The framework shows promise but needs additional experiments and rigorous evaluation to fully demonstrate its capabilities.

**Strengths:**

- First work to achieve joint generation of LiDAR point clouds and multi-view camera images with cross-modal consistency
- Novel epipolar-based cross-modal condition module addressing alignment between point clouds and images in different geometric spaces
- Multi-level controllable generation incorporating text, bounding boxes, images and point clouds as conditions

**Weaknesses:**

## Limitations in Innovation
- Weak theoretical foundation for cross-modal consistency, lacking rigorous proof of epipolar-based design effectiveness
- Incomplete solution for "spatial ambiguity", relying solely on epipolar geometry may not fully resolve 3D space ambiguities
- Limited innovation in generation control, mainly applying existing control methods to new scenarios

## Experimental Analysis

### Required Additional Experiments
- Evaluation of generated data utility in downstream tasks (3D object detection, trajectory prediction)
- Need to adopt more advanced transformer architectures like DiT instead of U-Net
- Consider replacing CLIP with T5 for potentially better performance
- Address low resolution issues affecting geometric shapes of distant objects (The current resolution is too low. Objects in the distance have no geometric shapes at all. However, the point cloud may have relatively complete and accurate geometric shapes in the distance. A larger resolution should be used, and because the generated image has slightly less geometric shapes in the distance, the effect is not good, simple and practical super-resolution model. It can only increase the resolution but cannot improve the distant geometry.)
- Investigate video generation capabilities with 3D VAE and 3D attention mechanisms
- Develop quantitative metrics for cross-modal consistency evaluation

## Ablation Study Analysis

### Missing Ablation Studies
- Detailed component-wise analysis of cross-modal epipolar condition module
- Impact assessment of different condition types on generation quality
- Sensitivity analysis of key hyperparameters
- Investigation of different training strategies

**Questions:**

While the work presents valuable theoretical contributions, significant improvements in code availability, documentation, and practical validation are needed for broader impact and reproducibility.

### 1. Code Release Improvements
- Release code on popular platforms (GitHub/GitLab)
- Provide comprehensive documentation
- Include example applications
- Add benchmark results reproduction scripts

---

> ### Author Response · Authors · 2024-11-22
> **Response to Reviewer SjU7**
>
> We appreciate your acknowledgment of our novel contribution about our problem setting, model design, and multi-level controllable generation. We try to address your concerns as follows.
>
> Q1. **Theoretical foundation for cross-modal consistency.**
>
> Our theoretical derivation is formulated with the following steps:
>
> (1) The multi-modality joint generation requires the **mutual cross-modality condition** in the denoising process of each modality (Eq.5).
>
> (2) The cross-modality alignment mainly relies on the condition on the **corresponding local information** from the other modality.
>
> (3) Our proposed epipolar-line cross-modality condition module can align the geometrical space (Eq.6,7) between LiDAR and camera modalities, which allows it to find the cross-modality local correspondence between LiDAR and camera feature maps.
> We agree that there is no rigorous mathematical proof, but it is consistent with intuition and should be enough to 1) formulate the multi-modality joint generation problem and 2) motivate the design of our epipolar-line cross-modality condition module. Our paper is more related to application instead of diffusion model theory, so the rigorous mathematical proof is out of our scope. This format is similar to previous ICLR/NeurIPS publications about multiview camera image generation with cross-view consistency claims [1,2,3].
>
> [1] MagicDrive: Street View Generation with Diverse 3D Geometry Control (ICLR 2024)
>
> [2] SyncDreamer: Generating Multiview-consistent Images from a Single-view Image (ICLR 2024)
>
> [3] Mvdream: Multi-view diffusion for 3d generation (NeurIPS 2023)
>
> Q2. **Epipolar geometry may not fully resolve 3D space ambiguities.**
>
> We agree that pure epipolar geometry may not fully solve the problem. However, diffusion models themselves also have strong ability to model the distribution of each single-modality data, especially the image diffusion model pretrained on large-scale datasets. Therefore, the epipolar-line cross-modality condition only needs to provide some extra meaningful guidance about the 3D spatial cues, which is enough to guide the generation of powerful diffusion models in an aligned manner for each modality. We show the performance gain from epipolar geometry in our ablation studies that it can significantly improve the cross-modality alignment.
>
> Q3. **Applying existing control methods.**
>
> This our main contribution focuses on multi-modality joint generation. We do not focus on fancy control signals different from prior works. In contrast, our contribution of controllable generation lies in the aligned control between images and point clouds. For example, in Fig.6, we can insert or delete objects simultaneously in the point clouds and multiview images, which cannot be achieved by previous works. It's worth mentioning that X-Drive is a flexible framework that can be easily combined with other single-modality controllable generation methods.
>
> Q4. **Data utility in downstream tasks.**
>
> We refer you to Reviewer vwWu Q3 where we show the usefulness of synthetic data in the training of multi-modality 3D object detection model.
>
> Q5. **Advanced transformer architectures/T5 text encoder/video generation.**
>
> Thanks for these useful suggestions! We position this work as the first attempt of LiDAR-camera joint generation, and we are willing to incorporate these advanced techniques in the future.
>
> Q6. **Higher resolution.**
>
> We follow the image resolution in MagicDrive, and the point clouds density is constrained by the LiDAR sensor in nuScenes dataset (32-beam). Our cross-modality condition is not the main bottleneck for increasing the resolution, so we can easily extend our framework to higher resolution following single-modality state-of-the-arts in the future.
>
> Q7. **Quantitative metric for cross-modality consistency.**
>
> Since there is no existing metric, we propose a novel DAS metric for this goal in the paper. We refer you to Sec. 5.1 and Appendix A for more details. Our results show the great gain in DAS metric brought by our X-Drive framework, verifying the ability of X-Drive to improve cross-modality consistency.
>
> Q8. **Extra ablation studies**.
>
> We refer you to Reviewer Tmsf Q2 for additional ablation studies, including the epipolar line component, two-stage training strategy, and quantitative sampling number along epipolar lines. Due to the limited time of rebuttal, we will include more ablation studies in the final version.
>
> Q9. **Code release.**
>
> We will release the code and all the documents upon the acceptance of this paper.
>
> Thanks again for your time and effort! For any other questions, please feel free to let us know during the rebuttal window.

---

> > ### Comment · Reviewer_SjU7 · 2024-11-26
> >
> > Thank you for your detailed response to my previous comments. After careful review of your rebuttal, I provide the following assessment:
> >
> > **Strengths:**
> > 1. Your response regarding theoretical foundations (Q1) effectively contextualizes your research within existing literature, particularly in relation to ICLR/NeurIPS publications.
> > 2. The explanation of the synergy between epipolar geometry and diffusion models (Q2) is convincing, well-supported by ablation studies.
> > 3. The proposed DAS metric shows innovation in evaluation methodology, though there remains room for improvement.
> >
> > **Areas for Improvement:**
> > 1. While your theoretical explanation aligns with an application-oriented paper, the final version would benefit from more rigorous mathematical derivations.
> > 2. Regarding resolution limitations (Q6): This inherited constraint from MagicDrive becomes more pronounced in the multi-modal context. Low resolution results in blurred or unrecognizable distant objects in images and sparse or missing points in point clouds. While increasing resolution demands more GPU memory, consider adopting memory optimization strategies from large language models.
> >
> > **Recommendations:**
> > 1. Include the additional ablation studies mentioned in Q8.
> > 2. Elaborate on potential integration with advanced Transformer architectures and resolution enhancement strategies.
> > 3. Add a section on limitations and future directions, emphasizing scalability aspects.
> > 4. Ensure timely code release.
> >
> > Maintaining a moderately positive rating of 6 points.

---

> ### Author Response · Authors · 2024-11-27
>
> We thank the reviewer for the detailed response. We are glad to see that our response can address your concerns. We will incorporate your suggestions in the final version and future work!

---

### Official Review · Reviewer_Tmsf · 2024-10-30

**Soundness:** 2
**Presentation:** 3
**Contribution:** 2
**Rating:** 5
**Confidence:** 4

**Summary:**

This work proposes X-Drive, explicitly addressing LiDAR-image consistency in driving data synthesis. It designs (1) mutual conditioning between corresponding local regions across modalities, (2) an epipolar line-based conditioning module, and (3) multi-source input conditions to mitigate generation inconsistencies. Experiments on nuScenes validate the efficacy of the proposed method.

**Strengths:**

- The focus on consistency between LiDAR and image generation provides insights to this field.

**Weaknesses:**

- Lack of quantitative comparison with baselines. In Table 1, the number of experiments conducted is insufficient to fully support this paper. The authors are encouraged to compare with more previous methods and implement additional vanilla baselines to better illustrate the advantages of X-Drive. Table 1 could be divided by modality and enriched separately.
- Lack of ablation studies. In Table 3, the ablation analysis is presented at a very coarse level. The authors are encouraged to detail the design of each module and analyze their individual impact on the final results.
- Marginal improvements. In Table 1, for JSD in the C→L setting, X-Drive only surpasses MagicDrive by 0.002, while RangeLDM surpasses LiDARGen by 0.106. For FID in the L→C setting, X-Drive only surpasses MagicDrive by 0.19, whereas MagicDrive surpasses BEVGen by 8.65.
- Numerous typos. For example, "several works emerges" in L103, "We extend the vanilla diffusion models to modeling" in L186-187, and "Our FID metric is outoerformed by MagicDrive" in L423-424. These errors should be reviewed and corrected to enhance the clarity and readability of the paper.

**Questions:**

- In Figure 1(b), what does the distribution at the bottom represent?
- In Figure 2, how is the generation of the LiDAR point cloud achieved?
- What are the additional time and memory costs of the designed module?
- What is the average generation time for one experiment?

---

> ### Author Response · Authors · 2024-11-22
> **Response to Reviewer Tmsf**
>
> We thank for your recognition of our insight, and we try to address your concerns as follows.
>
> Q1. **Lack of quantitative comparison with baselines.**
>
> Thanks for your suggestions. We will split the table in the final version. For image generation, we will include more baselines. For point clouds generation, **the three baselines in the table are the only ones reporting quantitative results on nuScenes dataset**, so we include them as baselines (We have to use nuScenes due to Reviewer wkMY Q2). For multi-modality generation, we can incorporate one more baseline that takes the same textual prompt and box conditions for point clouds and multiview image generation, which is exactly the "*w/o* cross-modality" item in our ablation studies (Tab.5). X-Drive can still notably outperform this vanilla baseline.
>
> Q2. **Lack of ablation studies.**
>
> Thanks for pointing it out. We include two additional ablation studies: (1) *w/o two-stage training* that directly trains the point clouds branch and multi-modality modules together by merging the second and third training stages. (2) *w/o epipolar line* that randomly sample the same number of points on the other modality feature map without epipolar line for cross-modality condition. (3) *half sampling points along epipolar lines* that reduces the number of sampled points along epipolar lines in the cross-modality condition module by half from [24, 24, 12, 6, 12, 24, 24] to [12, 12, 6, 3, 6, 12, 12] for each block.
>
> Our results show that the two-stage training can notably improve the quality of point clouds (JSD metric). The epipolar-line is important for the alignment between LiDAR and camera modalities (DAS metric). More sampling along epipolar lines are also critical for the reliance of cross-modality conditions (DAS metric). All ablations result in notable performance drops in comparison with our full model.
> | Methods | FID$\downarrow$ | JSD$\downarrow$ | DAS$\downarrow$ |
> | --- | ---| ---| --- |
> | *w/o* two-stage training | 23.65 | 0.113 | 1.84 |
> |*w/o* epipolar line | 25.35 | 0.074 | 2.23 |
> | half sampling points along epipolar lines | 22.31 | 0.072 | 1.93
> |full model | **20.17**| **0.070** | **1.67**|
>
>
> Q3. **Marginal improvements.**
>
> As we mentioned in Q2 of Reviewer vwWu, the goal of our paper is to enhance the cross-modality alignment instead of improving single-modality generation. In this case, we do not aim to significantly outperform prior work in the quality of each single-modality data. Our camera branch largely follows MagicDrive, and the LiDAR branch is similar with RangeLDM, since the single-modality models are not our focus. Thus, we only expect performance comparable with these prior works. We highlight the joint generation framework and cross-modality condition modules as our main contributions, which bring high cross-modality consistency reflected by the significant performance gain in DAS metric.
>
> Q4. **Typos.**
>
> Thanks for pointing them out. We will fix these typos and proofread again in the next version.
>
> Q5. **Distribution at the bottom of Fig.1(b).**
>
> This is just an illustration of our idea without quantitative referring. We want to show that our cross-modality condition can align the distributions of LiDAR point clouds and multiview camera images in the multi-modality generation to describe the same driving scene.
>
> Q6. **How is the generation of the LiDAR point cloud achieved?**
>
> After obtaining range-view representation with range and intensity $(r,i)$ for each coordinate $(\theta,\phi)$ in the range view, we can recover the corresponding point $(x,y,z)$ by inversing Eq.3 as follows.
> $z=r\sin(\phi),\ y=r\cos(\phi)\sin(\theta),\ x=r\cos(\phi)cos(\theta)$
> We will clarify it in the next version.
>
> Q7. **Latency of our method. Additional cost for the designed module.**
>
> We measure the latency for each module of our model. For the entire X-Drive model, it takes 7.2 seconds and 17GB GPU memory for the synthesis of each sample. For camera branch only, it takes 4.7 seconds and 12GB GPU memory. For LiDAR branch only, it takes 1.4 seconds and 4GB GPU memory. Thus, we think our cross-modality condition would not bring much extra cost.
>
> We hope our response will clarify the reviewer's confusion. If you have any questions, please feel free to let us know during the rebuttal window. And we sincerely hope to obtain support from the reviewer.

---

> ### Author Response · Authors · 2024-12-03
> **Discussion period will end soon**
>
> Dear Reviewer Tmsf,
>
> We would like to kindly remind you that the response period **will end within a few hours**, and we have not yet received your feedback. Given the importance of your feedback to us, we would greatly appreciate it if you could take the time to share your thoughts in the remaining hours.
>
> Thank you for your time and consideration, and we look forward to hearing from you.
>
> Best regards,
>
> The Authors

---

> > ### Author Response · Authors · 2024-12-03
> > **Kindly requesting you to consider raising your rating.**
> >
> > Dear Reviewer Tmsf,
> >
> > We greatly appreciate your constructive feedback. If our response and revision can help address your concerns, can we kindly request you consider raising your rating?
> >
> > Thank you very much for your time spent reviewing our paper. We are also glad to take any additional questions or suggestions.
> >
> > Best regards,
> >
> > Authors

---

### Official Review · Reviewer_vwWu · 2024-10-31

**Soundness:** 3
**Presentation:** 3
**Contribution:** 3
**Rating:** 6
**Confidence:** 4

**Summary:**

- This work proposes a novel technique to perform driving based lidar data generation in the form of range images, and multi-camera data generation while ensuring cross-modal alignment by leveraging latent diffusion models.
- Prior works address multi-camera image generation and lidar based generation individually while this work aims to perform joint generation of lidar and camera data. The major contribution here is in aligning the modalities during generation.
- During denoising of a first modality (camera or lidar), the information from the other modality (lidar or camera) is encoded and transformed into the geometrical space of the first modality (camera or lidar) to ensure alignment. They do this by finding matches from a second modality for a query from first modality by traversing the epipolar line in the second modality corresponding to the query.
- They show qualitative and quantitative results for conditional and unconditional generations. In addition, the authors also display controllability via bounding boxes and language interfaces.

**Strengths:**

- S1: The work addresses an important problem of lacking aligned lidar camera data. Solutions towards this can reduce/elimintate post-processing work of performing motion compensation via scene flow between consecutive pointcloud frames to address the lack of alignment. This is a cumbersome process.
- S2: The epipolar line based matching technique is neat, intuitive, and has been successfully been used before in literature [1].
- S3: The manuscript is well written and easy to follow. I also like the multi-view consistency results.

[1] Consistent View Synthesis with Pose-Guided Diffusion Models - https://arxiv.org/pdf/2303.17598

**Weaknesses:**

- W1: Mapping pointclouds to range images leads to information loss. It is understandable the authors use range images since most of the existing lidar generation work uses range images and in addition, image based generation techniques are much more mature than pointcloud counterparts. However, the major goal here is to ensure alignment between the modalities and not to perform unimodal generations which have been addressed before. Therefore, I would expect point clouds to be effective in filling gaps caused by information loss especially during occlusions and in far ranges.
- W2: I find the qualitative results of specifically the lidar generations lacking when viewed in the form of point clouds. Since the application of this work is to be used for training models, having distributional similarities (elaborated below) with real points is important to serve as valuable indictive biases and I doubt its utility without it.
    - Pattern realism: The lidar patterns shown in the pointcloud figures do not look realistic to me. For instance, the ground points on a rotating lidar typically show-up as thick concentric circles whereas results in supp. Fig 8 look very artificial to me. Comparing this to real pointclouds or even other synthetic lidar generation techniques from literature such as [2, 3], I see a gap that might be important to ignore for a downstream model.
    - Object realism: The point clouds also lack object shape or object realism in the figures presented in Fig 6 of main paper. For instance, vehicles have a typical shape based on their orientation which are useful biases for a downstream network that consumes this data. These patterns are very evident in other synthetic techniques [2, 3] but seems to be lacking here. Similarly, on the range images for the multi-modal generations in Fig 4, very prominent cars in the scene do not have as many lidar points as I would expect. This was an objective of the work as stated in the manuscript (L64) which I don’t believe has been fully fulfiled.
    - Following similar theme, the points on the pedestrian in Fig 4 do not seem realistic to me.
- W3: Related to W2, since the larger goal of this work is to generate data that can be used for training, I would have expected to see results showcasing this utility. The manuscript in the current form does not convey the effectiveness of the data towards this.
- W4: The authors state one of the application is to address long-tail scenarios (L52). They do show results in rain and bounding box can be a way to inject situtations such as cut-ins. However, these are mainly shown in the visual space currently which is also illustrated by prior works such as [4, 5]. The primary evaluation of the work proposed here is not in judging effectiveness of visual generation but alignment between cross-modal generations. The current results in the manuscript do not show such OOD scenarios which can be validated in the point cloud and image space.

I am weighting W2 > W1 > W3 = W4.

Overall, I like the work and the problem being tackled. However, I do not believe the results are quite there yet to be useful. Therefore, I am inclined to reject this work but I am open to changing my opinions based on the rebuttal and reviewer discussion.

Minor comments:

- Typo in L259 - Fourier
- L294 - I think you mean split vertically?
- Typo in L424.


References:

[2] Towards Realistic Scene Generation with LiDAR Diffusion Models - https://arxiv.org/pdf/2404.00815

[3] RangeLDM - https://arxiv.org/pdf/2403.10094

[4] MagicDrive - https://arxiv.org/pdf/2310.02601

[5] BEVGen - https://arxiv.org/pdf/2301.04634

[6] GenMM - https://arxiv.org/pdf/2406.10722

**Questions:**

- Q1: Baseline setting for Tab 1: I don’t completely understand how MagicDrive + RangeLDM was combined to obtain cross-modality score? Was there anything done to ensure alignment between the different modalities from the two models?
- Q2:  I was wondering if the authors made the decision of studying this problem in range-image space based on empirical results or out of convenience (prior work)? In other words, was denoising in pointcloud space studied for this problem?
- Q3: (suggestion) I wonder if the evaluation metrics are unable to capture objectness of pointclouds. Athough not typical in generative learning, techniques like Iterative closest point (ICP) could be useful in measuring shape discrepancies.
- Q4: Performing denoising in each modality while encoding information from the other modality through the denoising time steps seems expensive. It would be helpful to provide the computation time split between the different stages? If possible, also relate it to time taken for unimodal generations.
- Q5: Was the zero initialization from ControlNet deemed necessary empirically? I’m curious if the authors observed any forgetting without it.
- Q6: It looks like the results for lidar to camera is far better than the multimodal and camera to lidar results to me. Do the authors agree? If so, since generating point clouds is difficult, would disnetangling the joint process to a sequential approach of first generating camera data and then performing conditional lidar generation be feasible?
- Q7: Have the authors looked at GenMM [6]? While they do not specifically tackle multi-modal generations, they target alignment between the two modalities using in-painting diffusion models. I wonder on similar lines to Q5 if their approach can be conducive to a sequential approach of using reference patches from camera generations to condition generating point cloud data leveraging in-painting.

Prioritization order: Q1 > Q2 = Q4 > Q7 = Q6 > Q5 > Q3

I welcome the authors to correct me if there was any misunderstanding on my part.

---

> ### Author Response · Authors · 2024-11-22
> **Response to Reviewer vwWu [1/3]**
>
> We apprciate your recognition of our problem setting, model design, and paper writing. We try to address your concerns as follows:
>
> Q1. **Point clouds 3D representation or range-view representation.**
>
> We agree with you on the strengths of 3D point clouds representation. However, the goal of our work is to **synthesize sensor data** instead of reconstructing the 3D senes. We hope the synthetic point clouds can simulate the real data captured by the LiDAR sensor. If there are occlusion or distant sparsity in real LiDAR sensor data, we would expect the same patterns in synthetic data to avoid the domain gap between real and synthetic data. In this case, range-view representation is naturally compatible with the data collection process of LiDAR sensor (a unique point along each beam). Besides, since we focus on the cross-modality alignment instead of single-modality generation, we prefer to follow previous state-of-the-arts in point clouds generation. Our main contribution, the epipolar-line cross-modality condition, can be easily extended to 3D point clouds representation by sampling local features along the camera rays from the 3D point clouds representation.
>
> Q2. **Quality of point clouds generation.**
>
> We appreciate your careful observation. Before addressing your concrete concerns, we emphasize the following keypoints.
> 1. Previous point clouds generation works mainly report results for 64-beam LiDAR sensor (SemanticKITTI dataset) but we are working on 32-beam LiDAR point clouds (nuScenes data) with reasons explained in Reviewer wkMY Q2. The 64-beam data is naturally easier for model training and looks better due to its higher density. **For fairness, we only compare with 32-beam results of prior works,** which are only reported in LiDM (qualitatively in Fig.4 [2]) and RangeLDM (qualitatively and quantitatively in Fig.6 and Tab.4 [3]).
> 2. Our paper mainly focuses on **cross-modality alignment**. We roughly follow RangeLDM [3] in the design of point clouds branch except our extra box conditions. As a result, we expect a comparable rather than much better point clouds quality compared to RangeLDM **since we do not focus on notably improving the single-modality quality**.
> 3. Our multi-modality framework and epipolar-line cross-modality condition are flexibly compatible with various single-modality diffusion models since it only needs to sample points from feature maps. It can be easily combined with new single-modality state-of-the-arts in future.
>
> Additionally, we clarify the realism issues in the generated scenes as follows:
> - *Pattern realism:* In rainy weather, there should be some breaks in the concentric circles of LiDAR points (1st and 4th examples in Fig.8) due to the water on the ground, which is also witnessed in real point clouds (https://drive.google.com/file/d/1DTRSEKDvxX4UhAzgMkJgjJE2erwCEILQ/view?usp=sharing).
> This reflects the ability of our model in the generation of extreme weather. For sunny weather, the concentric circle patterns are also witnessed in point clouds generated by X-Drive (other examples in Fig.8).
> - *Object realism:* We show some qualitative comparison between X-Drive and RangeLDM/LiDM on 32-beam point clouds (https://drive.google.com/file/d/1DTRSEKDvxX4UhAzgMkJgjJE2erwCEILQ/view?usp=sharing).
> X-Drive shows comparable or even better object realism in comparison with LiDM and RangeLDM on 32-beam data. We agree that there are some gaps compared to real data for all these three methods, but our main goal is to improve the cross-modality consistency instead of single-modality point clouds quality. We will show in Q3 that current synthesis can already help downstream tasks.
> - *Pedestrian realism:* We acknowledge the improvement space for pedestrian point clouds synthesis, but this also exists in all existing works including LiDM and RangeLDM. Since the goal of our paper is not to improve the point clouds generation, we aim to achieve a single-modality performance comparable with previous work, but with a better cross-modality alignment of the entire scene.

---

> ### Author Response · Authors · 2024-11-22
> **Response to Reviewer vwWu [2/3]**
>
> Q3. **Can the synthesis help model training?**
>
> We conduct experiments on multi-sensor object detection with SparseFusion model [7]. We train a model with the combination of 1.4k real scenes (5% nuScenes training set) and 1.4k synethtic scenes, which is compared with a baseline trained only with 1.4k real scenes. Results show the helpful role of our synthetic data in the training of downstream multi-sensor perception models.
> | Training Data | mAP$\uparrow$ | NDS$\uparrow$ |
> | --- | ---| ---|
> | 1.4k real | 56.28| 51.54|
> |1.4k real + 1.4k synthetic | **57.60**| **52.23**|
>
> We also note that most existing point clouds generation works (like LiDM and RangeLDM) do not conduct the experiment about synthetic data for model training. As an orthogonal exploration of LiDAR-camera joint generation, we believe X-Drive can also benefit from the future development of single-modality algorithms to further boost the usefulness of synthetic data for downstream model training.
>
> [7] Xie, Yichen, et al. "Sparsefusion: Fusing multi-modal sparse representations for multi-sensor 3d object detection." ICCV 2023
>
> Q4. **Multi-modality OOD generation results.**
>
> Fig.6 shows the results of OOD generation including lighting, weather, object insertion and object removal. For object insertion and removal, we already show the results of both point clouds and multiview images. For lighting, LiDAR sensor would not be affected by lighting condition like daytime or night. For weather, rainy weather would interfere the relectance of laser, which is reflected by the sparse and messy point clouds shown in https://drive.google.com/file/d/1DTRSEKDvxX4UhAzgMkJgjJE2erwCEILQ/view?usp=sharing and also reflected in the 1st and 4th examples of Fig.8.
>
> Q5. **Baseline setting for RangeLDM+MagicDrive.**
>
> There is nothing guaranteeing the alignment between the different modalities from the two models RangeLDM and MagicDrive. However, we have no choice except this weak baseline **since RangeLDM is the only point clouds generation algorithm that we have access to either code or results on nuScenes dataset** (The authors kindly share with us the nuScenes generation results). As a stronger baseline, we refer you to the "*w/o* cross-modality" item in our ablation study Tab.3, where the point clouds and multiview images are generated independently based on the same bounding box and textual description conditions as a constraint of alignment. X-Drive also notably outperforms this strong baseline.
>
> Q6. **Decision of studying this problem in range-image space.**
>
> Please refer to our response to Q1.
>
> Q7. **ICP metric.**
>
> This is a very good suggestion. We will explore it in our future work.
>
> Q8. **Computational cost.**
>
> X-Drive does not require extra encoders except the light-weight epipolar-line module (Fig.3 or violet parts in Fig.2). The cross-modality condition is directly based on the feature maps from the diffusion models of each modality.
> We measure the latency and memory cost for each module of our model. For the entire X-Drive model, it takes 7.2 seconds and 17GB GPU memory for the synthesis of each sample. For the camera branch only, it takes 4.7 seconds and 12GB GPU memory. For the LiDAR branch only, it takes 1.4 seconds and 4GB GPU memory. Thus, we think our cross-modality condition would not bring much extra cost.
>
> Q9. **Effect of zero initialization.**
>
> We made a trial without it in the beginning. In the early stage of multi-modality training, the lack of zero initialization made the model converge very slowly and we also noticed the bad quality of each modality generative results, so we incorporated the zero-initialization strategy in all following experiments.

---

> ### Author Response · Authors · 2024-11-22
> **Response to Reviewer vwWu [3/3]**
>
> Q10. **Sequential image generation and then image-to-point-clouds generation.**
>
> This is feasible with our framework. People can first generate images guided by some conditions using other existing image generation algorithms like MagicDrive. Then, they can use the synthetic images as condition to generate point clouds through our camera-to-LiDAR conditional generation mode. This strategy can even allow more conditions supported by state-of-the-art image generation frameworks (like HD-map for MagicDrive), but it may lead to higher latency than joint multi-modality generation.
>
> Q11. **Discussion about GenMM.**
>
> Thanks for sharing this related work. They actually follow a sequential pipeline by firstly inpainting the video then inpainting the point clouds based on the video. However, their video-to-LiDAR step relies on some geometrical solutions, which is different from our diffusion model. The geometrical strategy has some limitations: Firstly, the precision relies on the high resolution of 3D volume, which makes it hard and very slow for from-scratch large-scale scene generation. Secondly, it is impossible to generate the reflectance intensity of LiDAR sensors.
>
> Q12. **Typos.**
>
> Thanks for pointing them out. We will fix these typos and proofread again in the next version.
>
> We wish that our response has addressed your concerns, and turns your assessment to the positive side. If you have any questions, please feel free to let us know during the rebuttal window. We appreciate your suggestions and comments!

---

> > ### Author Response · Authors · 2024-11-25
> > **Look forward to your response**
> >
> > Dear Reviewer vwWu,
> >
> > We would like to thank you very much for your insightful review, and we hope that our response addresses your previous concerns regarding our paper. However, as the discussion period is expected to end in the next few days, please feel free to let us know if you have any further comments on our work. We would be willing to address any additional concerns you may have. Otherwise, we hope that you will consider increasing your rating.
> >
> > Thank you again for spending time on the paper, we really appreciate it!
> >
> > Best regards,
> >
> > Authors

---

> > ### Comment · Reviewer_vwWu · 2024-11-27
> >
> > Thank you for the detailed rebuttal, I found it very helpful. Few follow-ups:
> >
> > - On pointcloud vs range-image: Not sure I understand exactly but if the goal is to synthesize sensor data, surely pointclouds are closer to the objective than range images? The LiDAR sensor data *is* in the form of pointclouds and range-image is obtained by post-processing these pointclouds. This operation leads to an irreversible information loss from occlusion and quantization etc. - typically I’ve seen this loss to be ~10% and they are therefore not 1:1.
> >     - Side note since I was pointed to wkMY Q2 - I believe Argoverse and Waymo datasets also contain LiDAR + multi-camera data.
> > - Qualitative evaluation:
> >     - I’m not sure if I am looking correctly but the objects in the shared file do not seem discernible to me (what objects are these and how were they generated?). So, I don’t know if I can judge X-Drive results being comparable/better with these visuals. I would expect a 32 channel LiDAR while not as dense as a 64 channel to still represent discernible object shapes. For example, Fig 6 in RangeLDM shows discernible vehicle shapes from a 32 channel LiDAR.
> >     - On pattern realism, I should’ve made it clear but my comment was about the rings not looking circular (not the breaks) in the synthetic pointcloud. This is unlike real pointclouds in my experience but perhaps that’s a weakness across all synthetic pointcloud techniques. Wondering if there was motion compensation performed on the points before converting to range image?
> >     - General note: I agree the goal is to improve consistency but also would like to highlight that as part of multi-modal consistency, this cannot be ignored completely. For instance, if the goal is to generate a scene with vehicles/pedestrians, the objects in the generated image should also have pointcloud representations at the appropriate locations that are representative of the object class, and pose. As pointed out before, this was also a claim in the manuscript (L62) and I do not see evidence for this.
> > - Downstream training - Thank you, this addresses W3.
> > - OOD: I agree rain leads to noisy pointclouds (typically in the near range closer to the sensor). Looking at the results in the attached file, I do see noise-like points also in the clear scene though which is unusual. However, I can see that the rainy scene is noisier especially around the host vehicle, so I am okay taking this as a mild positive signal.

---

> ### Author Response · Authors · 2024-11-28
> **Response to follow-ups**
>
> Dear Reviewer vwWu,
>
> Thank you for the detailed response. We are glad that our response can solve some of your concerns. We try to address your follow-ups as below:
>
> **Point clouds vs range-image**:
>
> Our goal is to synthesize sensor data. Unlike 3D object scans, LiDAR point clouds essentially construct a 2.5D scene from a single viewpoint instead of a full 3D point clouds [8,9]. The range image perfectly represents the LiDAR sweep in the spherical coordinate. Each pixel on the range image corresponds to a laser beam. **Since each beam would only be reflected at one certain position, there would only exist one unique value at each range image pixel.** The occlusions result from the laser beam reflection itself instead of either point clouds or range-image representations.
>
> In our implementation, since nuScenes provides point clouds, there is a minor loss in the conversion to range images. The vertical resolution of the range image is determined by the number of laser beams (32 for nuScenes), so there is no quantization loss in the vertical dimension. However, our horizontal resolution (1024) may not exactly match the rotation of the LiDAR sensor (about 1080 per sweep), which results in minor information loss due to horizontal quantization (about 5%).
>
> However, for point clouds representation, it is hard to mimic the 2.5D scene of LiDAR point clouds, i.e. **it is difficult to ensure one unique point along each laser beam**. We believe this would pose more challenges to the data synthesis than the minor quantization loss. We are trying to solve this challenge of point clouds representation in our future work.
>
>
> **Dataset with LiDAR + multi-camera data**:
>
> Thanks for pointing out the two datasets. For Waymo Open Dataset, they only provide front and side cameras **without rear view cameras for the ego-vehicle**, so there is a mismatch in the LiDAR and camera sensor data. For Argoverse, it also uses 32-beam LiDAR sensors and the dataset scale is similar with nuScenes, so it would not bring great difference compared to nuScenes. Since nuScenes is much more popular on various tasks, that's why we chose nuScenes as our dataset. We will include Argoverse as a future work.
>
>
> **Qualitative evaluation**:
>
> **Object realism:** We improve the layout of the [shared file](https://drive.google.com/file/d/1DTRSEKDvxX4UhAzgMkJgjJE2erwCEILQ/view?usp=sharing)) to make it more clear. In the shared file, we show the synthetic object point clouds from RangeLDM, LiDM, and X-Drive. We do not know the exact categories for RangeLDM and LiDM objects, since their models cannot handle object-level category condition. For X-Drive objects, we have denoted the object categories in the shared file. The 1st and 2nd object are distant, but their orientations are still clear. The 3rd and 4th objects clearly show the shapes and orientations of two trucks. The 5th object is clearly the left side of a car, where we can identify its overall contour.
>
> **Pattern realism:** Even in the real point clouds, the distant rings are not so circular (updated in the [shared file](https://drive.google.com/file/d/1DTRSEKDvxX4UhAzgMkJgjJE2erwCEILQ/view?usp=sharing)). It is possibly due to the lack of motion compensation in the same LiDAR sweep. Besides, the L2 loss of diffusion model is not sensitive to small position errors. This can be relieved by longer training schedules, but we cannot afford many computational resources in this part since we only have four GPUs.
>
> **General notes:** We agree with you in the importance of object-level quality, but these rely on **orthogonal** efforts. **The point clouds realism mainly relies on the point clouds branch** instead of the cross-modality condition. In our ablation studies, we show that our cross-modality condition would not hurt single-modality qualities (w/o cross-modality in Tab.3). At the time of our ICLR submission, there is no public code of point clouds generation model on nuScenes, so we have no choice but to develop the point clouds branch based on RangeLDM by ourselves which is trained in our first two training stages. For our main contribution, we develop **a plug-in cross-modality condition module** that can be injected between the feature maps of **any** SOTA multi-view images and LiDAR point clouds generative models. It can stand on the shoulders of single-modality giants due to the orthogonal nature. If someone releases a new SOTA point clouds generation model, we can easily replace our point clouds branch with their algorithm and delete our first two training stages, and all the point clouds realism issues can be solved naturally in this case.
>
>
> Thanks again for your time and efforts! Feel free to raise additional questions if you still have any concerns.
>
>
> [8] Meyer, Gregory P., et al. "Lasernet: An efficient probabilistic 3d object detector for autonomous driving." CVPR 2019.
>
> [9] Hu, Peiyun, et al. "What you see is what you get: Exploiting visibility for 3d object detection." CVPR 2020.

---

> > ### Comment · Reviewer_vwWu · 2024-11-28
> >
> > Thank you, the corresponding images in the shared file are helpful.
> >
> > (not necessary to respond, posting for clarification) Minor clarification regarding information loss - I was referring to the process of pointcloud -> range-image -> reconstructed pointcloud here where the reconstructed pointcloud will have fewer points than the original pointcloud. Since the objective is to synthesize sensor data, the reconstruction to pointcloud representation might be needed based on downstream use case.  The reconstructed pointcloud also would not have coverage for far-range points due to point collisions. I agree spherical transformation itself would not lead to any information loss.

---

> > > ### Author Response · Authors · 2024-12-02
> > > **Last day of the discussion period**
> > >
> > > Dear Reviewer vwWu,
> > >
> > > We appreciate your constructive feedbacks in the discussion period. Since today is the last day of the discussion period, can you kindly raise your score if our response is helpful to address your concerns?
> > >
> > > Thank you very much for your efforts in reviewing our paper.
> > >
> > > Best regards,
> > >
> > > Authors

---

> > > > ### Author Response · Authors · 2024-12-03
> > > > **Thank You**
> > > >
> > > > Dear Reviewer vwWu,
> > > >
> > > > We're glad our rebuttal addresses your concerns and appreciate that you increase your rating to 6.
> > > >
> > > > We will continue revising our paper based on constructive feedback from the reviewer and other reviewers. Please feel free to let us know if you have any further concerns.
> > > >
> > > > Best,
> > > >
> > > > Authors

---

> ### Author Response · Authors · 2024-11-28
> **Thanks for your response**
>
> Dear Reviewer vwWu,
>
> We are glad that our response can help. We agree that there is some inevitable loss in the conversion process. After observing the "point clouds -> range-image -> reconstructed point clouds" conversion process of ground-truth LiDAR data, we find it would lose about 5% of points, but it only has minor effect on the appearance of reconstructed point clouds. We will also explore the point clouds representation in our future work.
>
> We really appreciate your efforts in reviewing our paper and all the insightful feedbacks. If our response can address most of your concerns, can we humbly request you to raise your rating?
>
> Best regards,
>
> Authors

---

> > ### Author Response · Authors · 2024-11-30
> > **Kindly requesting you to consider raising your rating.**
> >
> > Dear Reviewer vwWu,
> >
> > We greatly appreciate your constructive feedbacks. If our response and revision can help to address your concerns, can we kindly request you to consider raising your rating?
> >
> > Thank you very much for your time spent on reviewing our paper. We are also glad to take any additional questions or suggestions.
> >
> > Best regards,
> >
> > Authors

---

### Official Review · Reviewer_wkMY · 2024-11-01

**Soundness:** 3
**Presentation:** 3
**Contribution:** 3
**Rating:** 6
**Confidence:** 4

**Summary:**

This paper introduces X-DRIVE, a novel framework for generating consistent multimodal data, specifically LiDAR point clouds and multi-view images, for autonomous driving scenarios. The key innovation of this paper is a cross-modality epipoplar condition module that bridges the geometrical gap under spatial ambiguity between point clouds and multi-view images, significantly enhancing modality consistency. Experimental results have demonstrated the effectiveness of the proposed method.

**Strengths:**

+ This paper introduces the first framework for the joint generation of LiDAR and camera data for autonomous driving scenarios.

+ The proposed cross-modality epipolar condition module to enhance modality consistency makes sense and works well.

+ The paper is well-written and easy to follow.

**Weaknesses:**

1. This paper presents a well-designed diffusion architecture for joint LiDAR and image data generation. My major concern is the novelty. This paper's main contribution is a cross-modality epipolar condition module for consistency constraints. However, epipolar attention for multi-view consistency has been extensively explored in multi-view diffusion models [1, 2, 3].

[1] Liu, Yuan, et al. "Syncdreamer: Generating multiview-consistent images from a single-view image." arXiv preprint arXiv:2309.03453 (2023).
[2] Shi, Yichun, et al. "Mvdream: Multi-view diffusion for 3d generation." arXiv preprint arXiv:2308.16512 (2023).
[3] Yang, Jiayu, et al. "Consistnet: Enforcing 3d consistency for multi-view images diffusion." Proceedings of the IEEE/CVF Conference on Computer Vision and Pattern Recognition. 2024.

2. Experiments are only conducted on one dataset, nuScenes.

3. It would be great to detail what kind of and how many GPUs are used, as well as the training/inference time.

**Questions:**

Is there a small mistake in Eq. (5)?

---

> ### Author Response · Authors · 2024-11-22
> **Response to Reviewer wkMY**
>
> We appreciate your recognition of our motivation, method design, and paper writing. We try to address your concerns as follows:
>
> Q1.**Existing application of epipolar line in multiview diffusion models.**
>
> Thank you for sharing these related works. However, their methods are largely different from our X-Drive. For [1,3], they construct an explicit 3D feature volume from multiview image features. This limits their methods to single object since feature volumes for large-scale outdoor scenes demand infeasible memory costs. In contrast, we do not need explicit 3D scene representation. For [2], it applies a joint global self-attention to aggregate multiview image features, which also has high memory cost, but we adopt an efficient local condition based on epipolar-line cross-attention.
>
> Besides, our contribution lies in the cross-modality alignment, which reflects a different motivation and insight compared with previous multiview diffusion model works. We will include these related works with discussions in our revision.
>
> [1] Liu, Yuan, et al. "Syncdreamer: Generating multiview-consistent images from a single-view image." arXiv preprint arXiv:2309.03453 (2023).
>
> [2] Shi, Yichun, et al. "Mvdream: Multi-view diffusion for 3d generation." arXiv preprint arXiv:2308.16512 (2023).
>
> [3] Yang, Jiayu, et al. "Consistnet: Enforcing 3d consistency for multi-view images diffusion." Proceedings of the IEEE/CVF Conference on Computer Vision and Pattern Recognition. 2024.
>
> Q2. **Experiments are only conducted on nuScenes dataset.**
>
> We conduct experiments on nuScenes dataset because it is the **only** mainstream dataset equipped with both **LiDAR sensor and multi-view surrounding cameras**. Such paired sensor data is necessary for our multi-modality experiments. In contrast, for other popular datasets, although they all have LiDAR data, SemanticKITTI only has front view cameras, KITTI360 uses fisheye cameras, and Waymo Open Dataset lacks back view cameras. We will extend our work to the robust case with missing sensors in future work.
>
> We also refer to previous publications about LiDAR-camera multi-modality 3D object detection. For the similar reason, most of them like [4,5,6] only provide experiments on nuScenes dataset.
>
> [4] Yang, Zeyu, et al. "Deepinteraction: 3d object detection via modality interaction." NeurIPS 2022
>
> [5] Xie, Yichen, et al. "Sparsefusion: Fusing multi-modal sparse representations for multi-sensor 3d object detection." ICCV 2023
>
> [6] Yan, Junjie, et al. "Cross modal transformer: Towards fast and robust 3d object detection." ICCV 2023
>
> Q3. **GPUs and training/inference time.**
>
> We are using four NVIDIA RTX A6000 GPUs for training, which is fewer than typical 8-16 GPUs for prior multi-view images or point clouds generation works. The first stage of training (point clouds VAE) takes about 1 day. The second stage (point clouds diffusion model) takes about 6 days. The third stage (multi-modality diffusion model) takes about 6 days. For inference, it takes about 7.2 seconds for each scene (point clouds + multiview images) with a single A6000 GPU.
>
> Q4. **Question about Eq.5.**
>
> This equation means the denoising of either LiDAR or camera modality is conditioned on both modalities. Could you specify your question if there is anything unclear? Thank you so much.
>
> Thanks again for your time and effort! For any other questions, please feel free to let us know during the rebuttal window.

---

> > ### Comment · Reviewer_wkMY · 2024-11-27
> >
> > Thank you the authors for the response.
> >
> > I don't have other questions.

---

> > > ### Author Response · Authors · 2024-11-27
> > > **Thank You**
> > >
> > > Dear Reviewer wkMY,
> > >
> > > We're glad our rebuttal addresses your concerns and appreciate that you maintain your rating of 6.
> > >
> > > We will continue revising our paper based on constructive feedback from the reviewer and other reviewers. Please feel free to let us know if you have any further concerns.
> > >
> > > Best,
> > >
> > > Authors

---

### Author Response · Authors · 2024-11-22
**General Response**

We sincerely appreciate all reviewers' time and efforts in reviewing our paper. We are glad to find that reviewers generally recognize our contributions including the meaningful problem setting and insight (all reviewers), the well-designed algorithm (Reviewer wkMY, vwWu, SjU7), the well-written paper (Reviewer wkMY, vwWu), as well as the experiments of multiview consistency (Reviewer vwWu) and controllable generation (Reviewer SjU7).

# Summary of new experiments
In response to reviewer comments, we have added several new experiments and results to strengthen the paper.
+ Exploiting synthetic multi-modality data to enhance the training of downstream multi-sensor 3D object detection model SparseFusion (Reviewer vwWu, SjU7).
    | Training Data | mAP$\uparrow$ | NDS$\uparrow$ |
    | --- | ---| ---|
    | 1.4k real | 56.28| 51.54|
    |1.4k real + 1.4k synthetic | **57.60**| **52.23**|
+ Qualitative comparison with prior SOTA single-modality point clouds generation algorithms (Reviewer vwWu). The anonymous links of these visualization comparisons are here: [Pattern Realism and Object Realism](https://drive.google.com/file/d/1DTRSEKDvxX4UhAzgMkJgjJE2erwCEILQ/view?usp=sharing).
+ Additional qualitative and quantitative ablation studies about the proposed cross-modality condition module (Reviewer Tmsf, SjU7).
    | Methods | FID$\downarrow$ | JSD$\downarrow$ | DAS$\downarrow$ |
    | --- | ---| ---| --- |
    | *w/o* two-stage training | 23.65 | 0.113 | 1.84 |
    |*w/o* epipolar line | 25.35 | 0.074 | 2.23 |
    | half sampling points along epipolar lines | 22.31 | 0.072 | 1.93
    |full model | **20.17**| **0.070** | **1.67**|
+ We provide the latency and cost analysis of our model.
---

The point-wise questions raised by reviewers are answered in the separate response to each reviewer. We thank all the reviewers for their constructive feedback which is helpful for the improvement of this paper. We hope that our responses can address current concerns and are happy to answer any further questions. We thank all reviewers' time again.

---

### Comment · Area_Chair_pVHZ · 2024-11-27
**Reminder: Last day for author feedback**

This is a reminder that today is the last day allotted for author feedback. If there are any more last minute comments, please send them by today.

---

### Author Response · Authors · 2024-11-28
**Summary of paper revisions**

Dear AC and reviewers,

We appreciate your efforts in reviewing our paper. Based on the suggestions of reviewers, we make the following revisions to enrich our paper:

1. We include the conversion from range image representation to point clouds representation in Sec. 3. (Reviewer Tmsf)
2. We split Tab. 1 based on the modality and incorporate two additional image generation baselines (WoVoGen [1] and LayoutDiffusion [2]). (Reviewer Tmsf)
3. We include Tab. 3 to show that our synthetic data can support the downstream perception model training. (Reviewer vwWu and SjU7)
4. We add the point clouds in rainy weather to Fig. 6. (Reviewer vwWu)
5. We re-organize ablation studies (Sec. 5.4) and incorporate three additional experiments in General Response into the paper. The ablation study of classifier-free guidance is moved to the appendix to save space.  (Reviewer Tmsf and SjU7)
6. We extend the future work section in the appendix based on the suggestions of reviewers. (Reviewer vwWu and SjU7)
7. We fix some typos and include some related works suggested by reviewers.
8. The appendix is attached to the end of main paper for easy reference.

We sincerely thank all reviewers for their constructive feedbacks which help to polish our paper. We are glad to take any additional questions during the discussion period.


Best regards,

Authors



[1] Lu, Jiachen, et al. "Wovogen: World volume-aware diffusion for controllable multi-camera driving scene generation." ECCV 2024.

[2] Zheng, Guangcong, et al. "Layoutdiffusion: Controllable diffusion model for layout-to-image generation." CVPR 2023.

---

### Meta-Review · Area_Chair_pVHZ · 2024-12-18

**Metareview:**

The paper proposes a method to synthesize consistent lidar and camera images for outdoor driving scenario using diffusion models. While it bares similarities to methods in single modality generation, the authors use an additional branch where the consistency between the two are enforced through the use of epipolar constraint/module. It also claims to be the first cross-modality generation method. We have read through the referee reports and the author responses. There are several common threads of concern involving epipolar constraint, realistic generation, and experiments (ablations, downstream applications, etc.). While some of the reviewers did not respond or confirm whether their questions or concerns were addressed, based on the author responses, we feel that they were mostly addressed. That said, the common concerns shared across reviewers included critical points, especially on experiments, and we recommend that revise the manuscript according to the reviewers' feedback.

**Additional Comments On Reviewer Discussion:**

Reviewer wkMY raised concerns regarding novelty as epipolar constraint has explore thoroughly explored. Reviewer vwWu had concerns regarding the realism of the generated results, choice of representation, and downstream applications. Reviewers Tmsf and SjU7 had concerns regarding utility, quantitative comparison with baselines and ablation studies. Additionally, SjU7 also raised concerns about whether the epipolar constraints are respected.

We believe wkMY's concerns were which were addressed by the authors. While vwWu did not confirm whether the authors addressed the concerns, based on the rebuttal, we feel that they were adequately addressed. Tmsf and SjU7 shared similar concerns, and the authors provided results, which addressed some of SjU7's concerns. While Tmsf did not respond, we feel that most of the questions and concerns were addressed.

---

### Decision · Program_Chairs · 2025-01-22

Accept (Poster)